# On the Benefits of Active Data Collection in Operator Learning

**Unique Subedi** [1]   **Ambuj Tewari** [1]

## Abstract

We study active data collection strategies for operator learning when the target operator is linear and the input functions are drawn from a mean-zero stochastic process with continuous covariance kernels. With an active data collection strategy, we establish an error convergence rate in terms of the decay rate of the eigenvalues of the covariance kernel. We can achieve arbitrarily fast error convergence rates with sufficiently rapid eigenvalue decay of the covariance kernels. This contrasts with the passive (i.i.d.) data collection strategies, where the convergence rate is never faster than linear decay ($\sim n^{-1}$). In fact, for our setting, we show a *non-vanishing* lower bound for any passive data collection strategy, regardless of the eigenvalues decay rate of the covariance kernel. Overall, our results show the benefit of active data collection strategies in operator learning over their passive counterparts.

## 1. Introduction

There is an increasing interest in using data-driven methods to estimate solution operators of partial differential equations (PDEs) encountered in scientific applications. To set up the problem, consider $\mathcal{X} \subseteq \mathbb{R}^d$ and a *linear* PDE of the form $\mathcal{L}u = f$, subject to the boundary condition $u(x) = 0$ for all $x \in \text{boundary}(\mathcal{X})$. The goal, given a function $f$, is to find the corresponding solution $u$ that satisfies the PDE.

Traditionally, numerical PDE solvers are used to compute $u$ from $f$. In contrast, operator learning focuses on approximating the solution operator of the PDE (Lu et al., 2021; Kovachki et al., 2023). Under typical conditions, this PDE has a linear solution operator $\mathcal{F}$ such that $u = \mathcal{F}(f)$ for all $f$ in an appropriate function space. Given a set of training samples $(f_1, u_1), \ldots, (f_n, u_n)$, the aim is to learn an operator $\widehat{\mathcal{F}}_n$ that closely approximates the true solution operator

$\mathcal{F}$ under a suitable metric. The hope is that for a new input function $f$, evaluating $\widehat{u} = \widehat{\mathcal{F}}_n(f)$ will be considerably faster than solving for $u$ through traditional methods, with minimal loss in accuracy.

In this work, we study the sample complexity of operator learning. Specifically, given an operator $\mathcal{F}$, how many input-output pairs $\{(f_j, \mathcal{F}(f_j))\}_{j=1}^n$ are necessary to estimate an operator $\widehat{\mathcal{F}}_n$ such that $\widehat{\mathcal{F}}_n(f) \approx \mathcal{F}(f)$ for all relevant $f$? This question has been studied in several specific contexts, such as for linear operators with fixed singular value decomposition by de Hoop et al. (2023) and Subedi & Tewari (2024), for Lipschitz operators by Liu et al. (2024), and for the random feature model by Nelsen & Stuart (2021). These are just a few representative works, and we refer the reader to (Kovachki et al., 2024b, Section 5) for a more comprehensive review of such sample complexity results.

A common theme of these sample complexity analyses is that they are conducted within the framework of traditional statistical learning (Kovachki et al., 2023, Section 2.2). In the statistical setting, the learner has access to training samples $\{(f_j, \mathcal{F}(f_j))\}_{j=1}^n$, where $f_j \sim_{\text{iid}} \mu$ from some probability measure $\mu$, and the objective is to produce an estimator $\widehat{\mathcal{F}}_n$ such that $\widehat{\mathcal{F}}_n(f) \approx \mathcal{F}(f)$ on average over test samples $f \sim \mu$. This scenario is also referred to as the passive learning setting. Under reasonable non-trivial assumptions, the best achievable rate of error convergence in this setting is $\sim 1/n$, when $\widehat{\mathcal{F}}_n$ is evaluated under an appropriate metric, say the $p$-th power of the $L_\mu^p$-Bochner norm.

### 1.1. Our Contribution

In this work, we go beyond the passive statistical setting and study operator learning where the learner is not restricted to iid samples from a source distribution and can use active data collection strategies. We focus on the case where the operator of interest $\mathcal{F}$ is a bounded *linear* operator and $\mu$ is a distribution with zero mean and the covariance structure defined by a continuous kernel $K$. Such distributions include Gaussian processes with common covariance kernels. For a given covariance kernel $K$, our main result provides an active data collection strategy, an estimation rule, and establishes an error bound for the proposed estimator in terms of the eigenvalue decay of the integral operator of $K$. Formally, if $\lambda_1 \geq \lambda_2 \geq \ldots$ are the eigenvalues of the

[1]Department of Statistics, University of Michigan, Ann Arbor, USA. Correspondence to: Unique Subedi <subedi@umich.edu>.

*Proceedings of the $42^{nd}$ International Conference on Machine Learning*, Vancouver, Canada. PMLR 267, 2025. Copyright 2025 by the author(s).

integral operator of $K$, there exists an active data collection strategy and estimation rule such that the estimator obtained using $n$ actively collected input-output pairs achieves the following error bound:

$$\varepsilon^2 \sum_{i=1}^{n} \lambda_i + \|\mathcal{F}\|_{\text{op}}^2 \sum_{i=n+1}^{\infty} \lambda_i.$$

The $\varepsilon^2$ captures the error of approximation oracle $\mathcal{O}$ for $\mathcal{F}$ that the learner has access to. For example, $\mathcal{O}$ could be the PDE solver used to generate training data for operator learning. Generally, $\varepsilon > 0$ is the irreducible error of the bound. The second term $O(\sum_{i>n} \lambda_i)$ is a reducible error, which goes to 0 as $n \to \infty$ for continuous kernels $K$ under the bounded domain. For example, for the covariance operator $\alpha(-\nabla^2 + \beta\mathbf{I})^{-\gamma}$ used by Li et al. (2021) and Kovachki et al. (2023), we show that the reducible error vanish at the rate $\lesssim n^{-\left(\frac{2\gamma}{d}-1\right)}$. Taking $2\gamma \gg d$, one can achieve *any* polynomial rate of decay. In fact, given any rate $R_n \to 0$ as $n \to \infty$, one can always construct a continuous kernel $K$ such that the reducible error decays faster than $R_n$. Thus, arbitrarily fast rates can be obtained using active data collection strategies. Our main result is formalized in Theorem 3.1, and the proof is based on the celebrated Karhunen–Loève decomposition for functions drawn from $\mu$ with covariance kernel $K$.

Furthermore, in Theorem 4.2, we show that, irrespective of the decay rate of the eigenvalues of the covariance kernel $K$, there always exists a bounded linear operator $\mathcal{F}$ and a distribution $\mu$ with covariance kernel $K$ such that the minimax estimation error fails to converge to 0 under *any* passive (i.i.d.) data collection strategy. In particular, for every $n$, even when $\varepsilon = 0$, we establish the minimax lower bound of

$$\|\mathcal{F}\|_{\text{op}}^2 \, \lambda_1$$

under any passive data collection strategy. That is, the lower bound does not vanish even as $n \to \infty$. Collectively, Theorems 3.1 and 4.2 establish a clear advantage of active data collection strategy for operator learning.

### 1.2. Related Works

Recent work by Musekamp et al. (2024) considers active methods for operator learning. However, in contrast to our approach of using linearity of the solution operator and the distributional family of interest, their methods rely on estimating uncertainty and identifying coreset. Additionally, their study is purely empirical and lacks theoretical guarantees. In a similar spirit, Li et al. (2024) study using active learning to select input functions from multiresolution datasets to lower the data cost.

On the theoretical side, a closely related work is by Kovachki et al. (2024a), who allow for active data collection

strategies. However, the upper bound in (Kovachki et al., 2024a, Theorem 3.3) is derived assuming that input functions $v_1, \ldots, v_n$ are drawn i.i.d. from $\mu$. Their proof, based on standard empirical risk minimization (ERM) analysis, achieves a convergence rate that, at best, matches the Monte Carlo rate of $n^{-1/2}$. Moreover, their lower bounds apply to both active and passive data collection strategies, suggesting that, for the nonparametric operator classes considered by Kovachki et al. (2024a), active learning provides no clear advantage over passive approaches. Exploring whether an adaptive data collection strategy, informed by the covariance of $\mu$ and targeting smaller subclasses within these broad nonparametric classes, could yield faster convergence rates remains an interesting direction for future research.

Additionally, Boullé et al. (2023) shares our objective of achieving faster convergence rates for PDEs with linear solution operators, but there are notable differences between their results and ours. First, their approach requires stronger control over the Hilbert-Schmidt norm of the operator $\mathcal{F}$, whereas we only require control over the operator norm. Notably, the Hilbert-Schmidt norm can be arbitrarily larger than the operator norm. Second, their estimator uses the specific structure of $\mathcal{F}$, particularly the Green's function, while we rely on black-box access to $\mathcal{F}$ via an $\varepsilon$-approximate oracle. Lastly, although both approaches introduces a term measuring the quality of training data ($\varepsilon$ in our bound and $\Gamma_\varepsilon$ in theirs), their definition of $\Gamma_\varepsilon$ is more technical and less intuitive. However, their guarantee is stronger, as their upper bound applies uniformly to any $L^2$-integrable input function, whereas our guarantees hold in expectation for inputs drawn from the distribution $\mu$.

Our work considers the setting where $\mu$ is defined by a stochastic process with a specific covariance structure. Such a $\mu$ was taken to be a Gaussian process with mean zero and covariance given by $\alpha(-\nabla^2 + \beta\mathbf{I})^{-\gamma}$ in (Bhattacharya et al., 2021; Li et al., 2021; Kovachki et al., 2023). The use of Karhunen–Loève decomposition for generating input functions is also discussed by Boullé & Townsend (2023, Section 4.1). Our upper bound also share conceptual similarities with results in (Lanthaler et al., 2022; Lanthaler, 2023), who established approximation error bounds, rather than estimation, in terms of s eigenvalues of covariance operator.

Finally, we highlight the ICML 2024 tutorial by Azizzadenesheli (2024), who mentions active data collection as an important future direction for operator learning. We also acknowledge the extensive literature on the learning-theoretic foundations of active learning (Settles, 2009). The active learning framework we adopt is known as the membership query model, which has a rich history in learning theory (Angluin, 1988). A more detailed discussion of various active learning models within the learning theory literature is deferred to Section 3.4.

## 2. Preliminaries

### 2.1. Notation

Let $\mathbb{R}, \mathbb{C}$ denote the set of real and complex numbers respectively. The set $\mathbb{N}$ and $\mathbb{Z}$ denote the natural numbers and integers. Define $\mathbb{N}_0 := \mathbb{N} \cup \{0\}$. For any $x \in \mathbb{R}^d$, we use $|x|_p$ to denote the $\ell^p$ norm of $x$. Given a set $\mathcal{X} \subseteq \mathbb{R}^d$, we use $L^2(\mathcal{X})$ to denote the space of squared integrable *real-valued* functions on $\mathcal{X}$ under some base measure $\nu$. For any $u \in L^2(\mathcal{X})$, we define $\|u\|_{L^2}^2 := \int_{\mathcal{X}} |u(x)|^2 \, d\nu(x)$. The notation $\nu$ is reserved for the base measure on $\mathcal{X}$, whereas $\mu$ will be used to denote the probability distribution over $L^2(\mathcal{X})$. For a linear operator $\mathcal{F} : L^2(\mathcal{X}) \to L^2(\mathcal{X})$, we define $\|\mathcal{F}\|_{\mathrm{op}} := \sup\{\|\mathcal{F}v\|_{L^2} : \|v\|_{L^2} = 1\}$. We use GP to denote Gaussian Process.

### 2.2. Distribution Over Function Space

Let $\mathcal{X} \subseteq \mathbb{R}^d$ be any compact set, $\mathcal{B}(\mathcal{X})$ denote the Borel sigma-algebra, and $\nu$ denote some finite measure on $\mathcal{X}$ (that is, $\nu(\mathcal{X}) < \infty$). Generally, we will take $\nu$ to be Lebesgue measure on $\mathcal{X}$ but sometimes it may be useful to take a weighted measure such as $\propto e^{-\alpha^2 |x|^2} \, dx$. Denote $L^2(\mathcal{X}, \mathcal{B}(\mathcal{X}), \nu)$ to be the set of all squared integrable functions on $\mathcal{X}$. From here on, we will drop the dependence on $\mathcal{B}(\mathcal{X})$ and $\nu$, and just write $L^2(\mathcal{X})$. Let $(\Omega, \Sigma, \mathbf{P})$ denote a probability space. We will consider a sequence of real-valued random variables $\{h_x : x \in \mathcal{X}\}$ defined over the probability space $(\Omega, \Sigma, \mathbf{P})$ that is centered, squared integrable, and has *continous* covariance kernel $K : \mathcal{X} \times \mathcal{X} \to \mathbb{R}$. Recall that covariance kernels are symmetric and positive definite. More precisely, for any $x, y \in \mathcal{X}$, the random variables $h_x$ satisfies

$$\mathbb{E}[h_x] = \int_{\Omega} h_x(\omega) \, d\mathbf{P}(\omega) = 0$$

$$\mathbb{E}[h_x^2] = \int_{\Omega} |h_x(\omega)|^2 \, d\mathbf{P}(\omega) < \infty$$

$$\mathbb{E}[h_x \, h_y] = \int_{\Omega} h_x(\omega) \, h_y(\omega) \, d\mathbf{P}(\omega) = K(y, x).$$

Next, we use this process to define a probability distribution over $L^2(\mathcal{X})$. To that end, it will be more convenient to write the process as a function $h : \mathcal{X} \times \Omega \to \mathbb{R}$. By definition, $h(x, \cdot)$ is $\Sigma$-measurable for every $x \in \mathcal{X}$. However, this is not enough to argue that $h$ is a random element of $L^2(\mathcal{X})$. Thus, to ensure measurability, we will only consider stochastic processes $h$ that satisfy the following: (i) The process $h$ is measurable with respect to product sigma algebra $\mathcal{B}(\mathcal{X}) \times \Sigma$ and (ii) For every $\omega \in \Omega$, the sample path $h(\cdot, \omega) : \mathcal{X} \to \mathbb{R}$ is an element of $L^2(\mathcal{X})$. Conditions (i) and (ii) ensure that $\omega \mapsto h(\cdot, \omega)$ is a measurable function from $\Omega$ to $L^2(\mathcal{X})$ (Hsing & Eubank, 2015, Theorem 7.4.1). In other words, $h$ is a $L^2(\mathcal{X})$ valued random variable. We

can now meaningfully talk about probability distribution over $L^2(\mathcal{X})$ induced by the stochastic process $h$.

Accordingly, given a continuous covariance kernel $K$, let $\mathcal{P}(K)$ denote the set of all centered and squared-integrable stochastic processes with covariance kernel $K$ indexed by $\mathcal{X}$ that satisfies conditions (i) and (ii) above. With a slight abuse of notation, we will also use $\mathcal{P}(K)$ to denote the set of all distributions over $L^2(\mathcal{X})$ induced by these stochastic processes. Each element $\mu \in \mathcal{P}(K)$ is now a probability distribution over $L^2(\mathcal{X})$.

### 2.3. Problem Setting and Goal

Let $\mathcal{F} : L^2(\mathcal{X}) \to L^2(\mathcal{X})$ denote the operator of interest. One should think of $\mathcal{F}$ as the solution operator of the PDE. The goal is to estimate a surrogate $\widehat{\mathcal{F}}_n$ using $n$ input/output functions such that

$$\sup_{\mu \in \mathcal{P}(K)} \mathbb{E}_{v \sim \mu} \left[ \left\| \widehat{\mathcal{F}}_n(v) - \mathcal{F}(v) \right\|_{L^2}^2 \right] \tag{1}$$

is small. In the absence of additional knowledge about the image space of the solution operator $\mathcal{F}$, minimizing this objective is the most natural choice. Accordingly, for a fixed $\mu$, the $L_\mu^p$-Bochner norm has been a standard error metric in the operator learning literature (see (Kovachki et al., 2023, Section 2.2), (Liu et al., 2024)). The $p = 2$ case, in particular, is of practical significance, as its empirical counterpart is the widely used mean squared loss. Regarding the family of probability distributions, our proposed family $\mathcal{P}(K)$ aims to unify and generalize marginal distributions on input functions commonly used in practice (Li et al., 2021; Lu et al., 2021). This family also aligns with the recommendation of Boullé & Townsend (2023). Other families of probability distributions, such as the set of all compactly supported measures on a Hilbert space, have been used in theoretical analyses (e.g., (Liu et al., 2024)). Extending our result to include other distribution families of theoretical or applied interest is left for future work.

Throughout this work, we will assume that the learner knows the covariance kernel $K$.

**Assumption 2.1.** The learner knows the kernel $K$.

Although not always explicitly stated, this has been a standard assumption in the operator learning literature. For example, Li et al. (2021) and Kovachki et al. (2023) generate their input functions, both during training and testing, from a Gaussian process with the covariance kernel $K$ such that its associated integral operator is $\alpha(-\nabla^2 + \beta \mathbf{I})^{-\gamma}$ for some constants $\alpha, \beta, \gamma > 0$. Thus, all the empirical performances observed in these works are in a setup similar to those described above. Additionally, Boullé & Townsend (2023, Section 4.1.1) also suggests generating source terms (input functions) from Gaussian processes with standard covariance kernels such as RBF, Mattern, etc.

Additionally, from a learning-theoretic perspective, assuming knowledge of the kernel $K$ is arguably without loss of generality. In active learning, it is common to assume access to an unlimited pool of unlabeled samples $v_1, \ldots, v_m \sim_{\text{iid}} \mu$, where $\mu \in \mathcal{P}(K)$, and focus on minimizing label complexity—the number of labeled samples requested (Hanneke, 2013). This aligns with our setting, where labeling (e.g., solving a PDE) is the primary cost. Given such unlabeled samples, one can estimate the covariance operator as

$$\Sigma_m = \frac{1}{m-1} \sum_{i=1}^{m} (v_i - \bar{v}_m) \otimes (v_i - \bar{v}_m)$$

where $\bar{v}_m = \frac{1}{m} \sum_{i=1}^{m} v_i$. Since $\mathbb{E}[\|v_i\|^2] < \infty$, Theorem 8.1.2 of (Hsing & Eubank, 2015) guarantees that $\Sigma_m \to \Sigma$ almost surely in Hilbert-Schmidt norm, where $\Sigma$ is the integral operator associated with $K$. While our work assumes $\Sigma$ has a finite trace norm, this is not required to recover its eigenfunctions: convergence in Hilbert-Schmidt norm suffices for accurate spectral approximation. Thus, assuming access to the eigenfunctions of $K$ is reasonable in theory, even if it may be computationally demanding in practice.

Once the input functions are generated, the learner has to use numerical solvers to PDE numerically in order to generate the solution function. In this work, we will make the following assumption about learner's access to the PDE solver.

**Assumption 2.2.** The learner only has black-box access to $\mathcal{F}$ through an $\varepsilon$-approximate oracle $\mathcal{O}$ that satisfies

$$\sup_{v \in L^2(\mathcal{X})} \|\mathcal{O}(v) - \mathcal{F}(v)\|_{L^2}^2 \leq \varepsilon^2.$$

From an implementation standpoint, it might seem unnatural to consider $\mathcal{O}(v)$ for a function $v \in L^2(\mathcal{X})$, especially since most PDE solvers usually only take function values over a discrete grid as an input. Nevertheless, the oracle is an abstract object, and the grid can be integrated into its definition. For example, given any function $v$, the oracle first extracts the values of $v$ on a grid $\{x_1, \ldots, x_m\}$ and produces output values on the same or a different grid. On the output side, the oracle may then construct an actual function, either through trigonometric interpolation or simply by setting the function values to zero outside the grid points. Thus, we do not specify these implementation details of the oracle and instead characterize it solely by accuracy parameter $\varepsilon$.

In general, $\varepsilon$ primarily reflects the discretization error for finite-difference type methods and truncation error for spectral methods, but it may also include measurement errors or errors resulting from the early stopping of some iterative routine. Therefore, $\varepsilon$ can be broadly viewed as quantifying the quality of the training data. From this perspective, $\varepsilon$ represents the irreducible error in (1). Specifically, there exists

a function $g : [0, \infty) \to [0, \infty)$ such that (1) is bounded below by $g(\varepsilon)$, even as $n \to \infty$. There is extensive literature that attempts to quantify $\varepsilon$ for various oracles (PDE solvers), and we can use these results readily to establish bounds on the irreducible error in our context. For example, for spectral solvers truncated to the first $N$ basis functions where the input and output functions are $s$-times continuously differentiable, we typically have $\varepsilon \sim N^{-s/d}$. Here, $d$ is the dimension of the domain $\Omega$.

## 3. Upper Bounds Under Active Data Collection

In Section 2.3, we discussed the problem setting and the goal. Next, we specify how the learner can collect the training data $(v_1, w_1), \ldots, (v_n, w_n)$. In a departure from the standard statistical learning setting, where the training data is obtained as iid samples from the distribution under which the learner is evaluated, we investigate active data collection strategies. In active data collection strategies, the learner can pick *any* source terms $v_1, \ldots, v_n$ and use the oracle to obtain $w_i = \mathcal{O}(v_i)$. Since the goal is to provide guarantees under samples from the distribution $\mu \in \mathcal{P}(K)$, the learner *can* use the knowledge of $K$ to pick source terms. For a given oracle with accuracy $\varepsilon$, covariance kernel $K$, and the desired accuracy $\eta > 0$, the goal of the learner is to develop an active data collection strategy for the source terms and an estimation rule to produce $\widehat{\mathcal{F}}$ such that the accuracy of $\eta$ can be obtained with the fewest number of oracle calls. Or equivalently, achieve an optimal decay in the upperbound of (1) for $n \in \mathbb{N}$ number of oracle calls. Under this model, we provide an upperbound on (1) when $\mathcal{F}$ is a bounded linear operator.

**Theorem 3.1** (Upper Bound). *Suppose $\mathcal{F}$ is a bounded linear operator. There exists a deterministic data collection strategy and a deterministic estimation rule such that the estimate $\widehat{\mathcal{F}}_n$ produced after $n$ calls to oracle $\mathcal{O}$ satisfies*

$$\sup_{\mu \in \mathcal{P}(K)} \mathbb{E}_{v \sim \mu} \left[ \left\| \widehat{\mathcal{F}}_n(v) - \mathcal{F}(v) \right\|_{L^2}^2 \right] \leq \varepsilon^2 \sum_{i=1}^{n} \lambda_i + \|\mathcal{F}\|_{\text{op}}^2 \sum_{i>n} \lambda_i.$$

*Here, $\lambda_1 \geq \lambda_2 \geq \ldots$ are the eigenvalues of the integral operator of $K$ defined as $(\mathcal{I}_K v)(\cdot) = \int_{\mathcal{X}} K(\cdot, x) \, v(x) \, d\nu(x)$.*

The first term above is the irreducible error, which depends on the quality of the training data. For the second term, Hsing & Eubank (2015, Theorem 4.6.7) implies that

$$\sum_{i=1}^{\infty} \lambda_i = \int_{\mathcal{X}} K(x, x) \, d\nu(x) \leq \sup_x |K(x, x)| \, \nu(\mathcal{X}) < \infty.$$

This is finite because $\nu$ is a finite measure on $\mathcal{X}$, and $K(x, x)$ is a continuous function on a compact domain, making it bounded. As a result, the second term in the upper bound of Theorem 3.1 vanishes as $n \to \infty$. In Section 3.3, we apply Theorem 3.1 to derive precise rates for several common covariance kernels.

## 3.1. Data Collection Strategy and The Estimator

Here, we specify the data collection strategy and the estimator that achieves the claimed guarantee in Theorem 3.1. Let $\{\lambda_j, \varphi_j\}_{j=1}^{\infty}$ be the sequence of eigenpairs of $K$ defined by solving the Feldholm integral equation

$$\int_{\mathcal{X}} K(y, x)\, \varphi_j(x)\, d\nu(x) = \lambda_j\, \varphi_j(y), \qquad y, x \in \mathcal{X}.$$

Given the Oracle call budget of $n$, the input functions that the learner selects are $\varphi_1, \varphi_2, \ldots, \varphi_n$ as source terms. For each $i \in [n]$, the learner makes an oracle call and obtains $w_i = \mathcal{O}(\varphi_i)$. Then, we consider the estimator

$$\widehat{\mathcal{F}}_n := \sum_{i=1}^{n} w_i \otimes \varphi_i.$$

More precisely, this estimation rule yields an operator $\widehat{\mathcal{F}}_n$ such that $\widehat{\mathcal{F}}_n v = \sum_{i=1}^{n} w_i \langle \varphi_i, v \rangle_{L^2}$ for any $v \in L^2(\mathcal{X})$. Appendix A.1 provides an overview of the process for deriving this estimator starting from a least-squares estimation rule. Furthermore, Appendix C discusses methods for approximating the eigenfunctions $\varphi_i$ when the Fredholm integral equation cannot be solved exactly.

## 3.2. Sketch of a Proof of Theorem 3.1

We now provide a high-level, non-rigorous sketch of a proof of Theorem 3.1, and defer a full proof to Appendix A.

To bound the risk of the estimator specified above, we first rewrite the risk using Karhunen–Loève Theorem. Pick any $v \sim \mu$. Since $v$ is defined using a centered and squared-integrable stochastic process with continuous covariance kernel $K$, the celebrated Karhunen–Loève Theorem (Hsing & Eubank, 2015, Theorem 7.3.5) states that

$$v(\cdot) = \sum_{j=1}^{\infty} \sqrt{\lambda_j}\, \xi_j\, \varphi_j(\cdot),$$

where $\xi_j$'s are random variables defined as $\xi_j := \frac{1}{\sqrt{\lambda_j}} \int_{\mathcal{X}} v(x)\, \varphi_j(x)\, d\nu(x)$. It turns out that $\xi_j$'s are uncorrelated random variables with mean $0$ and variance $1$. That is, $\mathbb{E}[\xi_j] = 0$ and $\mathbb{E}[\xi_i\, \xi_j] = \mathbb{1}[i = j]$.

This decomposition allows us to rewrite expectation over $\mu$ in terms of expectation over the randomness of the sequence $(\xi_j)_{j \geq 1}$, which is more tractable. For simplicity, assume that $\varepsilon = 0$. Then, using Karhunen–Loève expansion, we can show that

$$\mathbb{E}_{v \sim \mu}\left[ \left\| \widehat{\mathcal{F}}_n(v) - \mathcal{F}(v) \right\|_{L^2}^2 \right] \leq \mathbb{E}_{\xi}\left[ \left\| \mathcal{F}\left( \sum_{j>n} \sqrt{\lambda_j}\, \xi_j\, \varphi_j \right) \right\|_{L^2}^2 \right],$$

which can then be further upper bounded by $\|\mathcal{F}\|_{\text{op}}^2 \sum_{j>n} \lambda_j$ using properties of $\xi_i$'s.

There are two primary challenges in completing this argument in a fully rigorous manner. First, we must address the fact that the oracle $\mathcal{O}$ is only $\varepsilon$-approximate for any $\varepsilon > 0$. Second, the convergence statement for $\sum_{j=1}^{\infty} \sqrt{\lambda_j} \xi_j \varphi_j(\cdot)$ is quite specific, requiring careful attention when applying this result.

## 3.3. Examples of Covariance Kernels

To make the upperbound in Theorem 3.1 more concrete, let us consider a few specific covariance kernels $K$. While not all claims are rigorously proven in this subsection, a detailed and formal treatment of the material can be found in Appendix B.

### 3.3.1. FRACTIONAL INVERSE OF SHIFTED LAPLACIAN

Li et al. (2021); Kovachki et al. (2023) generated input functions from $\text{GP}(0, \alpha(-\nabla^2 + \beta \mathbf{I})^{-\gamma})$ for some constants $\alpha, \beta, \gamma > 0$. Here, $\nabla^2$ is the Laplacian operator defined as

$$\nabla^2 v = \sum_{j=1}^{d} \frac{\partial^2 v}{\partial x_j^2}.$$

In this section, we will consider $\mathcal{X}$ to be a $d$-dimensional periodic torus $\mathbb{T}^d$ and the base measure $\nu$ is Lebesgue. We identify $\mathbb{T}^d$ by $[0, 1]^d$ with periodic boundary conditions. Let us define a function $\varphi_m : \mathbb{T}^d \to \mathbb{C}$ as $\varphi_m(x) = e^{2\pi\, \mathrm{i}\, m \cdot x}$ for every $m \in \mathbb{Z}^d$. Recall that $\varphi_m$ is the eigenfunction of $\nabla^2$ with eigenvalue $-4\pi^2 |m|_2^2$. In particular,

$$\nabla^2 e^{2\pi\, \mathrm{i}\, m \cdot x} = \sum_{j=1}^{d} \frac{\partial^2}{\partial x_j^2} e^{2\pi\, \mathrm{i}\, m \cdot x} = -4\pi^2 |m|_2^2\, e^{2\pi\, \mathrm{i}\, m \cdot x}.$$

Since $\{\varphi_m : m \in \mathbb{Z}^d\}$ forms a complete orthonormal system in $L^2(\mathbb{T}^d)$, there are no other eigenfunctions of $\nabla^2$. A simple algebra shows that $\varphi_m$'s are also the eigenfunctions of $(-\nabla^2 + \beta \mathbf{I})^{-\gamma}$ with eigenvalues being $(\beta + 4\pi^2 |m|_2^2)^{-\gamma}$.

Using this fact in Theorem 3.1 yields the upper bound

$$\leq \varepsilon^2 \left( \alpha \beta^{-\gamma} + \alpha + \frac{\alpha}{2\gamma - d} \right) + \frac{\alpha \|\mathcal{F}\|_{\text{op}}^2}{2\gamma - d}\, \frac{1}{n^{\frac{2\gamma}{d} - 1}}.$$

When $2\gamma/d - 1 > 0$, the reducible error above goes to $0$ when $n \to \infty$. Again, as an example, Li et al. (2021) uses $\alpha = 7^{3/2}$, $\beta = 49$ and $\gamma = 2.5$ in their experiment for $2d$-Navier Stokes. In this case, $2\gamma/d = 2.5$, yielding the convergence rate of $n^{-1.5}$ for the reducible error. Note that this rate is faster than the usual passive statistical rate of $1/n$. However, for any value $\tau$, one can take $\gamma = d(\tau + 1)/2$ to get the rate of $n^{-\tau}$. Thus, every polynomial rate is possible depending on the choice of $\gamma$.

### 3.3.2. RBF KERNEL

Let $\mathcal{X} = \mathbb{R}$ and $K(x, y) = \exp\left(-\frac{1}{2\ell^2}|x - y|^2\right)$ for all $x, y \in \mathbb{R}$ and $\ell > 0$. For now, let $\nu$ is a Gaussian measure with mean 0 and variance $\sigma^2$ on $\mathbb{R}$. Using the known results on eigenfunctions of RBF kernel in terms of Hermite polynomials (Williams & Rasmussen, 2006, Section 4.3.1), we show that there exists $\gamma \in (0, 1)$ such that the upper bound in Theorem 3.1 is

$$\leq \frac{1}{(1 - \gamma)}\left(\varepsilon^2 + \|\mathcal{F}\|_{\text{op}}^2 \gamma^n\right).$$

That is, the reducible error vanishes exponentially fast as $n \to \infty$. In Appendix B, we also show that a rate faster than any polynomial rate can be achieved for RBF kernel on $\mathbb{R}^d$.

### 3.3.3. BROWNIAN MOTION

Let us consider the case where $\mathcal{X} = [0, 1]$, the base measure $\nu$ is Lebegsue, and the stochastic process in Section 2.2 is Brownian motion. Recall that the Brownian motion is a Gaussian process with covariance kernel $K(s, t) = \min(s, t)$ for all $s, t \in [0, 1]$. It is well-known (Hsing & Eubank, 2015, Example 4.6.3) that the eigenfunctions of $K$ can be written in terms of sine waves. A simple analysis can then be used to establish an upper bound of

$$\leq \frac{\varepsilon^2}{2} + \|\mathcal{F}\|_{\text{op}}^2 \frac{1}{\pi^2} \frac{2}{2n - 1}.$$

Therefore, the reducible error vanishes at rate $\sim n^{-1}$.

### 3.4. Comparison to Traditional Active Learning

The active learning framework we adopt in this work is referred to as the membership query model, which has a long-standing history in the learning theory literature (Angluin, 1988; 2001). However, in traditional learning settings, the membership query model—where the learner can request labels for any unlabeled instance—is generally unrealistic. For example, in the context of human data, it may not be feasible to generate a label for an individual with an arbitrary feature vector, as such a person may not exist in reality. As a result, other active learning frameworks, such as the stream-based sampling model (Atlas et al., 1989) and the pool-based model (Lewis & Gale, 1994; Hanneke, 2013), have gained prominence in the recent literature. These models restrict the learner to requesting labels for instances sampled from a specific distribution, making them more practical for many real-world applications. For a comprehensive review of active learning models, their history, and key results, we refer readers to (Settles, 2009). That said, we believe that the membership query model is the right model for developing surrogates for solution operators of PDEs. This is because a PDE solver can provide a solution to any

query of an input function within an appropriate function space.

## 4. Lower Bounds on Passive Learning

In this section, we establish a lower bound on (1) for any passive data collection strategy. As usual, the kernel $K$ is known to the learner. Nature selects a distribution $\mu_\star \in \mathcal{P}(K)$, and the learner receives $n$ i.i.d. samples $v_1, v_2, \ldots, v_n \sim \mu_\star$. For each $i \in [n]$, the learner queries the oracle $\mathcal{O}$ to produce $w_i = \mathcal{O}(v_i)$. It is important to emphasize that the learner can only make oracle calls for the i.i.d. samples $v_1, v_2, \ldots, v_n$. If the learner were allowed to make oracle calls for other input functions, the learner could simply disregard these i.i.d. samples and implement the "active strategy" from Section 3.1. Such restriction on oracle calls still includes most passive learning rules of interest, such as arbitrary empirical risk minimization (ERM), regularized least-squares estimators, and parametric operators trained with stochastic gradient descent.

Using these $n$ training points $\{(v_i, w_i)\}_{i \leq n}$, the learner then constructs an operator $\widehat{\mathcal{F}}_n$. Since the learner only has access to samples from $\mu_\star$, it is unrealistic to expect a uniform guarantee over the entire family $\mathcal{P}(K)$ as established in Theorem 3.1. Therefore, in this section, the learner will be evaluated solely under the distribution $\mu_\star$. The objective is to minimize the expected loss under $\mu_\star$, defined as

$$\mathbb{E}_{v_{1:n} \sim \mu_\star^n}\left[\mathbb{E}_{v \sim \mu_\star}\left[\left\|\widehat{\mathcal{F}}_n(v) - \mathcal{F}(v)\right\|_{L^2}^2\right]\right].$$

Moreover, establishing any meaningful lower bound on this risk requires imposing some restriction on the oracle $\mathcal{O}$. To understand why, consider the case where $\mathcal{F}$ is a finite-rank operator that only maps to the span of $\{\psi_1, \psi_2, \ldots, \psi_N\}$ for some orthonormal sequence $\psi_1, \ldots, \psi_N$ in $L^2(\mathcal{X})$. Now, consider an oracle $\mathcal{O}$ such that for any $v \in L^2(\mathcal{X})$, it outputs

$$\mathcal{O}(v) = \mathcal{F}(v) + \chi\,\psi_{N+1},$$

where $\chi \in \mathbb{R}$ and $\psi_{N+1}$ is a unit norm function in $L^2(\mathcal{X})$ that is orthogonal to all $\psi_j$ for $1 \leq j \leq N$. If $|\chi| \leq \varepsilon$, it is easy to see that

$$\sup_{v \in L^2(\mathcal{X})} \|\mathcal{O}(v) - \mathcal{F}(v)\|_{L^2}^2 = \|\chi\,\psi_{N+1}\|_{L^2}^2 = |\chi|^2 \leq \varepsilon^2.$$

Thus, $\mathcal{O}$ is a valid oracle according to Assumption 2.2. However, in principle, it is possible to encode the entire identity of $\mathcal{F}$ in a real number $\chi$. Thus, the learner could determine the identity of $\mathcal{F}$ with just a single call to $\mathcal{O}$, making any attempt at establishing a lower bound futile.

This problem may still persist even when $\varepsilon = 0$. Consider the case where $\nu$ is the Lebesgue measure, and the oracle is

of the form

$$\mathcal{O}(v) = \mathcal{F}(v) + \chi \mathbb{1}[x = x_0]$$

for some $x_0 \in \mathcal{X}$. Then, for any $v \in L^2(\mathcal{X})$, we have $\|\mathcal{O}(v) - \mathcal{F}(v)\|_{L^2}^2 = \|\chi \mathbb{1}\{x = x_0\}\|_{L^2}^2 = 0$ as $\nu(\{x_0\}) = 0$. This shows that the oracle can still reveal the identity of $\mathcal{F}$ in regions of the domain that have zero measure under $\nu$. Therefore, to avoid these pathological edge cases, we will assume that the oracle is perfect.

**Definition 4.1** (Perfect Oracle). $\mathcal{O}$ is a perfect oracle for $\mathcal{F}$ if, for every $v \in L^2(\mathcal{X})$, we have

$$\big(\mathcal{O}(v)\big)(x) = \big(\mathcal{F}(v)\big)(x) \quad \forall x \in \mathcal{X}.$$

In other words, the perfect oracle $\mathcal{O}$ produces exactly the same function that $\mathcal{F}$ does—nothing more, nothing less. With this assumption, we are in the usual realizable setting often considered in statistical learning theory. That is, the learner has access to $n$ samples $\{(v_i, \mathcal{F}(v_i))\}_{i=1}^n$, where $v_1, \ldots, v_n$ are drawn iid from some distribution $\mu$.

Theorem 4.2 provides a lower bound on the risk of any estimator under such passive data collection strategy.

**Theorem 4.2** (Lowerbound). *Fix any continuous covariance kernel $K$ with eigenvalues $\lambda_1 \geq \lambda_2 \geq \ldots$. Then, there exists a solution operator $\mathcal{F}$, accessible to the learner through a perfect oracle $\mathcal{O}$, such that the following holds: for every $n \in \mathbb{N}$, there exists a distribution $\mu \in \mathcal{P}(K)$ such that, under any estimation rule within a passive data collection strategy, the risk of the resulting estimator $\widehat{\mathcal{F}}_n$ is*

$$\mathop{\mathbb{E}}_{v_{1:n} \sim \mu^n} \left[ \mathop{\mathbb{E}}_{v \sim \mu} \left[ \left\| \widehat{\mathcal{F}}_n(v) - \mathcal{F}(v) \right\|_{L^2}^2 \right] \right] \geq \frac{\|\mathcal{F}\|_{\text{op}}^2}{2} \sum_{j=1}^m \lambda_j$$

*for every fixed $m \in \mathbb{N}$.*

Specifically, for $m = 1$, we obtain a lower bound of $\frac{1}{2} \|\mathcal{F}\|_{\text{op}}^2 \lambda_1$. This provides a non-vanishing lower bound for any non-trivial operator $\mathcal{F}$ and covariance kernel $K$.

Our lower bound is constructive: we explicitly define a difficult distribution for the learner. We construct a distribution $\mu$ over input functions such that, along each eigenfunction direction $\varphi_j$, it places mass 0 with probability $1 - p$, and $\pm 1/\sqrt{p}$ with equal probability $p/2$. This yields a sparse distribution with rare but large spikes. A careful argument shows that this construction defines a valid distribution in $\mathcal{P}(K)$ for any $p > 0$. When $p$ is small, the learner observes mostly zero inputs during training with probability at least $1/2$, yet the expected squared error along each direction is $(1/\sqrt{p})^2 \cdot p = 1$, leading to a non-vanishing error. The full proof is provided in Appendix D.

# 5. Experiments

In this section, we conduct numerical studies comparing our active data collection strategy with passive data collection (random sampling) for learning solution operators for the Poisson and Heat Equations. For the actively collected data, we implement the *linear estimator* defined in Section 3.1. On the other hand, for passively collected data, we use a least-squares estimator, where the pseudoinverse is computed numerically. Recall that, given input-output functions $\{v_i, w_i\}_{i=1}^n$, the least-squares estimator has a form $L = (\sum_{i=1}^n w_i \otimes v_i)(\sum_{i=1}^n v_i \otimes v_i)^\dagger$. For the actively collected data in Section 3.1, the $v_i$'s are orthogonal, which yielded a simple and natural pseudoinverse (see Appendix A.1). However, for the passively collected data, the $v_i$'s may not be orthogonal anymore and the pseudoinverse does not have a nice closed form. Thus, we use standard numerical techniques to compute the pseudo-inverse $(\sum_{i=1}^n v_i \otimes v_i)^\dagger$. However, in practice, one rarely uses linear estimators for passively collected data. Thus, we also compare our method against the Fourier Neural Operator (Li et al., 2021), the most popular architecture for operator learning. Our code is available at `https://github.com/unique-subedi/active-operator-learning`.

## 5.1. Poisson Equation

Let $\mathcal{X} = [0, 1]^2$. Consider Poisson equation with Dirichlet boundary conditions:

$$-\nabla^2 u = f, \quad u(x) = 0 \quad \forall x \in \text{boundary}(\mathcal{X}),$$

where $\nabla^2$ is the Laplace operator. The objective is to learn the solution operator that maps the source function $f$ to the solution $u$. This solution operator is the inverse of the Laplacian, which is a compact linear operator since $\mathcal{X}$ is bounded. For the passive data collection strategy, the input functions $f$ are independently sampled as $f \sim \text{GP}(0, 50^2(-\nabla^2 + \mathbf{I})^{-2})$, where GP denotes Gaussian Process.

The solution $u$ is computed using the finite-difference method. Both linear estimators and Fourier Neural Operators (FNO) are trained on $n$ such independently sampled pairs $(f, u)$. For testing, 100 additional source functions $f \sim \text{GP}(0, 50^2(-\nabla^2 + \mathbf{I})^{-2})$ are generated, and their corresponding solutions $u$ are also obtained via the finite-difference method. Both active and passive estimators are evaluated on this test set, with the performance measured using the mean-squared relative error:

$$\text{Error} = \frac{1}{n_{\text{test}}} \sum_{i=1}^{n_{\text{test}}} \frac{\left\| u_i^{\text{true}} - u_i^{\text{predicted}} \right\|_{L^2}^2}{\|u_i^{\text{true}}\|_{L^2}^2}.$$

We report the relative error instead of the absolute error to normalize for potential arbitrary scaling due to the norms

of the true solution function. The FNO model has four Fourier layers and $N/2$ Fourier modes, where $N$ denotes the number of grid points along each spatial dimension. In our experiments, all computations are carried out on a $64 \times 64$ grid, so $N = 64$. Figures (1) and (2) show the testing error as a function of the training sample size. The performance of FNO on active data is not included in this figure due to its poor results. However, Figure (5) in the Appendix includes the error curve for FNO trained with active data, alongside the results for passive data.

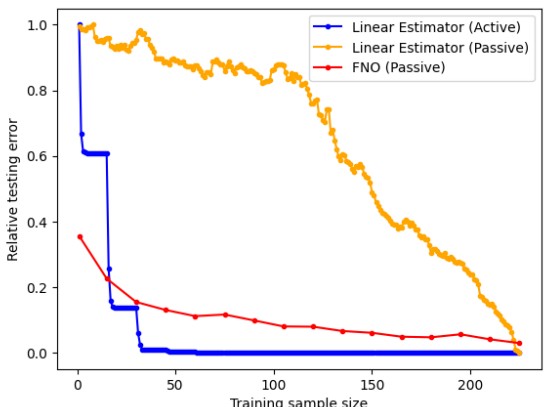

Figure 1. Error Plots for various estimators for Poisson Equation. The blue curve shows the performance of our linear estimator on actively collected data. The orange and red curves include the linear estimator's and FNO's performance on passively collected data. Figure 2 shows the same plot in log-scale.

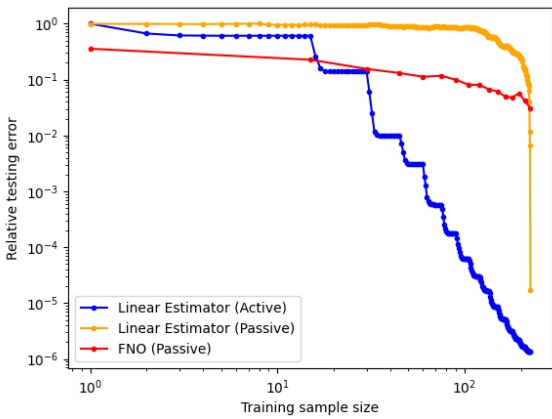

Figure 2. Error Plots for Poisson Equation in log-log scale.

While the convergence guarantee of our estimator is formally established only for the covariance operator $50^2(-\nabla^2 + \mathbf{I})^{-\gamma}$ with $\gamma > 1$ as $d = 2$, we observe that the estimator demonstrates robust convergence even when $\gamma \leq 1$ in the context of Poisson equation. Figure (3) presents

the convergence rate of our estimator in log scale, using actively collected data across various values of $\gamma$.

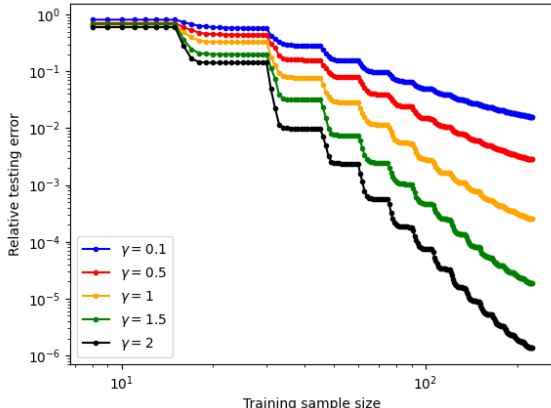

Figure 3. Convergence rate of the active linear estimator for Poisson equation with actively collected data for different values of $\gamma$.

### 5.2. Heat Equation

Consider the heat equation

$$\frac{\partial u}{\partial t} = \tau \, \nabla^2 u,$$

where $u : [0,1]^2 \to \mathbb{R}$ vanishes on the boundary. The solution operator for this equation is given by $\exp(\tau t \nabla^2)$, and the solution at time $t \geq 0$ can be expressed as $u_t = \exp(\tau t \nabla^2)u_0$. Fixing $t = 1$, our objective is to learn the solution operator $\exp(\tau \nabla^2)$. This operator is defined as

$$\exp(\tau \nabla^2) = \sum_{k=0}^{\infty} \frac{(\tau \nabla^2)^k}{k!},$$

which is a bounded linear operator. As in the previous case, we sample $n$ initial conditions $u_0 \sim \mathrm{GP}(0, (-\nabla^2 + \mathbf{I})^{-1.5})$. For each initial condition, we use the finite difference method with forward-time discretization to compute the solution $u_1$ at $t = 1$. This is done using 1000 time discretization steps on a $64 \times 64$ grid. For our experiments, we set $\tau = 10^{-2}$. As $\tau$ is the step size in the forward Euler method, choosing a larger $\tau$ would result in instability in the numerical PDE solver.

All estimators are evaluated on a test set of size 100, drawn from the same distribution as the training data. Figure (4) presents the relative testing errors. Furthermore, the error plot for the Fourier Neural Operator (FNO) trained on actively collected data is shown in Figure (6) in the Appendix. Finally, Figure (7) in the Appendix shows the convergence rates of the active linear estimator for different values of $\gamma$.

Our experimental results verify the theoretical advantage of active data collection strategies over passive sampling, as

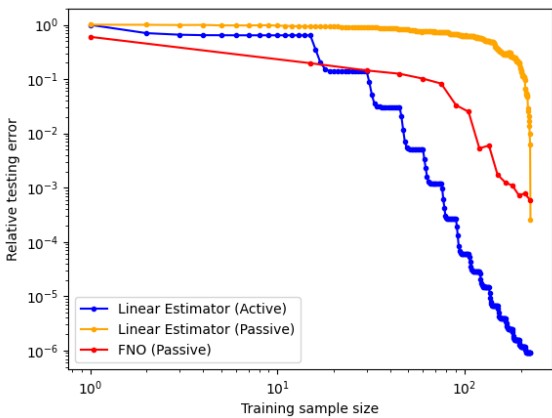

*Figure 4.* Error Plots for the Heat equation in log-scale.

established in Theorem 3.1. These findings highlight the practical utility of active learning frameworks in improving data efficiency for operator learning tasks.

## 6. Discussion and Future Work

In this work, we show that arbitrarily fast rates can be achieved with an active data collection strategy when the operator of interest is a bounded linear operator and the input functions are drawn from centered distributions with continuous covariance kernels. A natural extension of these results would involve non-linear operators. Specifically, one might ask whether there exists a natural class of non-linear operators that permits such fast rates when input functions are drawn from centered distributions with continuous covariance kernels. A natural starting point might be to consider the RKHS of operators. Additionally, given that functional PCA is the estimation of truncated Karhunen–Loève decomposition, it would be interesting to explore whether a variant of a PCANet-based architecture could achieve fast rates with active data collection.

## Impact Statement

This paper presents work whose goal is to advance the field of Machine Learning for scientific applications. There are many potential societal consequences of our work, none of which we feel must be specifically highlighted here.

## Acknowledgements

We acknowledge the support of NSF via grant DMS-2413089.

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

# A. Proof of Theorem 3.1

### A.1. Specifying Data Collection Strategy and The Estimator

We first specify the estimator that achieves the claimed guarantee in Theorem 3.1. Let $\{\lambda_j, \varphi_j\}_{j=1}^{\infty}$ be the sequence of eigenpairs of $K$ defined by solving the Feldholm integral equation

$$\int_{\mathcal{X}} K(y, x)\, \varphi_j(x)\, d\nu(x) = \lambda_j\, \varphi_j(y), \qquad y, x \in \mathcal{X}.$$

Given the Oracle call budget of $n$, the input functions that the learner selects are $\varphi_1, \varphi_2, \ldots, \varphi_n$ as source terms. For each $i \in [n]$, the learner makes an oracle call and obtain

$$w_i = \mathcal{O}(\varphi_i).$$

Consider the estimation rule

$$\underset{L \text{ is linear}}{\arg\min} \sum_{i=1}^{n} \|L\varphi_i - w_i\|_{L^2}^2 \,.$$

Solving this optimization problem boils down to solving the linear equation

$$\sum_{i=1}^{n} w_i \otimes \varphi_i = L \circ \left( \sum_{i=1}^{n} \varphi_i \otimes \varphi_i \right).$$

It is clear that this system is ill-posed and has infinitely many solutions. The family of solutions can be written as

$$L = \left( \sum_{i=1}^{n} w_i \otimes \varphi_i \right) \left( \sum_{i=1}^{n} \varphi_i \otimes \varphi_i \right)^{\dagger},$$

where $\dagger$ indicates the pseudoinverse. Each particular choice of pseudoinverse yields a distinct solution. Since $\varphi_i$'s are orthonormal, a natural one is

$$\left( \sum_{i=1}^{n} \varphi_i \otimes \varphi_i \right)^{\dagger} = \sum_{i=1}^{n} \varphi_i \otimes \varphi_i.$$

This choice of pseudoinverse yields the estimator

$$\widehat{\mathcal{F}}_n := \sum_{i=1}^{n} w_i \otimes \varphi_i,$$

which will be our estimator interest.

### A.2. Rewriting Risk using Karhunen–Loève Theorem

Next, we bound the risk of this estimator. Pick any $v \sim \mu$. Since $v$ is defined using a centered and squared-integrable stochastic process with continuous covariance kernel $K$, the celebrated Karhunen–Loève Theorem (Hsing & Eubank, 2015, Theorem 7.3.5) states that

$$v(\cdot) = \sum_{j=1}^{\infty} \sqrt{\lambda_j}\, \xi_j\, \varphi_j(\cdot),$$

where $\xi_j$'s are random variables defined as

$$\xi_j := \frac{1}{\sqrt{\lambda_j}} \int_{\mathcal{X}} v_x(\omega)\, \varphi_j(x)\, d\nu(x).$$

Here, $v_x(\omega)$ is simply just $v(x)$, but we write the dependence on $\omega$ explicitly to highlight the fact that $v$ is generated by stochastic process on the probability space $(\Omega, \Sigma, \mathbf{P})$.

It turns out that $\xi_j$'s are uncorrelated random variables with mean 0 and variance 1. In particular, we have

$$\mathbb{E}[\xi_j] = 0 \quad \text{and} \quad \mathbb{E}[\xi_i\,\xi_j] = \mathbb{1}[i = j].$$

The precise convergence statement is

$$\lim_{m\to\infty} \sup_{x\in\mathcal{X}} \mathbb{E}\left[\left|v(x) - \sum_{j=1}^{m} \sqrt{\lambda_j}\,\xi_j\varphi_j(x)\right|^2\right] = 0. \tag{2}$$

We refer the reader to standard texts (Hsing & Eubank, 2015, Theorem 7.3.5) or (Lord et al., 2014, Theorem 7.52) for the full proof of Karhunen–Loève Theorem.

Fix $m \in \mathbb{N}$ such that $m > n$ and define $\Pi_m$ to be a projection operator onto the first $m$ eigenfunctions of $K$. That is, for each $v$ with Karhunen–Loève decomposition $v(\cdot) = \sum_{j=1}^{\infty} \sqrt{\lambda_j}\,\xi_j\,\varphi_j(\cdot)$, we define

$$\Pi_m(v) := \sum_{j=1}^{m} \sqrt{\lambda_j}\,\xi_j\varphi_j(\cdot).$$

Since both $\widehat{\mathcal{F}}_n$ and $\mathcal{F}$ are linear operators, we can write

$$\mathbb{E}_{v\sim\mu}\left[\left\|\widehat{\mathcal{F}}_n(v) - \mathcal{F}(v)\right\|_{L^2}^2\right]$$

$$= \mathbb{E}_{v\sim\mu}\left[\left\|\widehat{\mathcal{F}}_n\big(\Pi_m(v)\big) - \mathcal{F}\big(\Pi_m(v)\big) + \big(\widehat{\mathcal{F}}_n - \mathcal{F}\big)\big(v - \Pi_m(v)\big)\right\|_{L^2}^2\right]$$

$$\leq \mathbb{E}_{v\sim\mu}\left[\left\|\widehat{\mathcal{F}}_n\big(\Pi_m(v)\big) - \mathcal{F}\big(\Pi_m(v)\big)\right\|_{L^2}^2\right] + 2\,\mathbb{E}\left[\left\|\widehat{\mathcal{F}}_n\big(\Pi_m(v)\big) - \mathcal{F}\big(\Pi_m(v)\big)\right\|_{L^2}\left\|\big(\widehat{\mathcal{F}}_n - \mathcal{F}\big)\big(v - \Pi_m(v)\big)\right\|_{L^2}\right]$$

$$+ \mathbb{E}\left[\left\|\big(\widehat{\mathcal{F}}_n - \mathcal{F}\big)\big(v - \Pi_m(v)\big)\right\|_{L^2}^2\right]$$

The inequality follows upon using triangle inequality and expanding the square. For the cross term, Cauchy–Schwarz inequality yields

$$\mathbb{E}\left[\left\|\widehat{\mathcal{F}}_n\big(\Pi_m(v)\big) - \mathcal{F}\big(\Pi_m(v)\big)\right\|_{L^2}\left\|\big(\widehat{\mathcal{F}}_n - \mathcal{F}\big)\big(v - \Pi_m(v)\big)\right\|_{L^2}\right]$$

$$\leq \sqrt{\mathbb{E}\left[\left\|\widehat{\mathcal{F}}_n\big(\Pi_m(v)\big) - \mathcal{F}\big(\Pi_m(v)\big)\right\|_{L^2}^2\right]} \sqrt{\mathbb{E}\left[\left\|\big(\widehat{\mathcal{F}}_n - \mathcal{F}\big)\big(v - \Pi_m(v)\big)\right\|_{L^2}^2\right]}.$$

Thus, we can write

$$\mathbb{E}_{v\sim\mu}\left[\left\|\widehat{\mathcal{F}}_n(v) - \mathcal{F}(v)\right\|_{L^2}^2\right] \leq (\mathrm{I}) + 2\sqrt{(\mathrm{I})\,(\mathrm{II})} + (\mathrm{II}),$$

where we define

$$(\mathrm{I}) := \mathbb{E}_{v\sim\mu}\left[\left\|\widehat{\mathcal{F}}_n\big(\Pi_m(v)\big) - \mathcal{F}\big(\Pi_m(v)\big)\right\|_{L^2}^2\right]$$

$$(\mathrm{II}) := \mathbb{E}\left[\left\|\big(\widehat{\mathcal{F}}_n - \mathcal{F}\big)\big(v - \Pi_m(v)\big)\right\|_{L^2}^2\right].$$

Next, we will bound (I) and (II) separately.

### A.3. Bounding (I).

Pick any $v \sim \mu$. Then, we know that there exists $\{\xi_j\}_{j\in\mathbb{N}}$ such that $v = \sum_{j=1}^{\infty} \sqrt{\lambda_j}\,\xi_j\varphi_j$. So, $\Pi_m(v) = \sum_{j=1}^{m} \sqrt{\lambda_j}\,\xi_j\varphi_j$, which subsequently implies that

$$\widehat{\mathcal{F}}_n\big(\Pi_m(v)\big) = \widehat{\mathcal{F}}_n\left(\sum_{j=1}^{m} \sqrt{\lambda_j}\,\xi_j\,\varphi_j\right) = \sum_{j=1}^{m} \sqrt{\lambda_j}\,\xi_j\,\widehat{\mathcal{F}}_n(\varphi_j) = \sum_{j=1}^{n} \sqrt{\lambda_j}\,\xi_j\,w_j,$$

where the final equality uses the fact that $n < m$ and $\widehat{\mathcal{F}}_n(\varphi_j) = 0$ for all $j > n$. Defining $\delta_i := \mathcal{O}(\varphi_i) - \mathcal{F}(\varphi_i)$, we obtain $w_i = \mathcal{F}(\varphi_i) + \delta_i$. This allows us to write

$$
\begin{aligned}
\widehat{\mathcal{F}}_n\big(\Pi_m(v)\big) &= \sum_{i=1}^{n} \sqrt{\lambda_i}\, \xi_i \left(\mathcal{F}(\varphi_i) + \delta_i\right) \\
&= \mathcal{F}\left(\sum_{i=1}^{n} \sqrt{\lambda_i}\, \xi_i \varphi_i\right) + \sum_{i=1}^{n} \sqrt{\lambda_i}\, \xi_i \delta_i \\
&= \mathcal{F}\left(\Pi_m(v)\right) - \mathcal{F}\left(\sum_{j=n+1}^{m} \sqrt{\lambda_j}\, \xi_j \varphi_j\right) + \sum_{i=1}^{n} \sqrt{\lambda_i}\, \xi_i \delta_i.
\end{aligned}
$$

So, we can rewrite (I) as

$$
\begin{aligned}
&\mathbb{E}_{v \sim \mu}\left[\left\|\widehat{\mathcal{F}}_n\big(\Pi_m(v)\big) - \mathcal{F}\big(\Pi_m(v)\big)\right\|_{L^2}^2\right] \\
&= \mathbb{E}_{\xi}\left[\left\|\sum_{i=1}^{n} \sqrt{\lambda_i}\, \xi_i\, \delta_i - \mathcal{F}\left(\sum_{j=n+1}^{m} \sqrt{\lambda_j}\xi_j\, \varphi_j\right)\right\|_{L^2}^2\right] \\
&= \mathbb{E}_{\xi}\left[\left\|\sum_{i=1}^{n} \sqrt{\lambda_i}\, \xi_i\, \delta_i\right\|_{L^2}^2\right] - 2\,\mathbb{E}_{\xi}\left[\left\langle \sum_{i=1}^{n} \sqrt{\lambda_i}\, \xi_i\, \delta_i, \mathcal{F}\left(\sum_{j=n+1}^{m} \sqrt{\lambda_j}\xi_j\, \varphi_j\right)\right\rangle\right] + \mathbb{E}_{\xi}\left[\left\|\mathcal{F}\left(\sum_{j=n+1}^{m} \sqrt{\lambda_j}\, \xi_j\, \varphi_j\right)\right\|_{L^2}^2\right].
\end{aligned}
$$

The cross-term vanishes upon swapping sum and integral as $\xi_i$'s are zero mean and uncorrelated. For the first term, note that

$$
\begin{aligned}
\mathbb{E}_{\xi}\left[\left\|\sum_{i=1}^{n} \sqrt{\lambda_i}\, \xi_i\, \delta_i\right\|_{L^2}^2\right] &= \mathbb{E}_{\xi}\left[\left\langle \sum_{i=1}^{n} \sqrt{\lambda_i}\, \xi_i\, \delta_i, \sum_{i=1}^{n} \sqrt{\lambda_i}\, \xi_i\, \delta_i\right\rangle_{L^2}\right] \\
&= \sum_{i=1}^{n} \lambda_i\, \mathbb{E}[\xi_i^2]\, \|\delta_i\|_{L^2} + 2 \sum_{1 \le i < j \le n} \sqrt{\lambda_i \lambda_j}\, \mathbb{E}[\xi_i \xi_j]\, \langle \delta_i, \delta_j\rangle_{L^2} \\
&= \sum_{i=1}^{n} \lambda_i\, \|\delta_i\|_{L^2}^2 + 0 \\
&\le \varepsilon^2 \sum_{i=1}^{n} \lambda_i.
\end{aligned}
$$

The final inequality uses the fact that $\mathcal{O}$ is $\varepsilon$-approximate for $\mathcal{F}$. For the third term, similar arguments show that

$$
\begin{aligned}
\mathbb{E}_{\xi}\left[\left\|\mathcal{F}\left(\sum_{j=n+1}^{m} \sqrt{\lambda_j}\, \xi_j\, \varphi_j\right)\right\|_{L^2}^2\right] &\le \|\mathcal{F}\|_{\mathrm{op}}^2\, \mathbb{E}_{\xi}\left[\left\|\sum_{j=n+1}^{m} \sqrt{\lambda_j}\, \xi_j\, \varphi_j\right\|_{L^2}^2\right] \\
&= \|\mathcal{F}\|_{\mathrm{op}}^2 \sum_{j=n+1}^{m} \lambda_j\, \|\varphi_j\|_{L^2}^2 \\
&= \|\mathcal{F}\|_{\mathrm{op}}^2 \sum_{j=n+1}^{m} \lambda_j.
\end{aligned}
$$

Thus, we have established that

$$
\text{(I)} \le \varepsilon^2 \sum_{i=1}^{n} \lambda_i + \|\mathcal{F}\|_{\mathrm{op}}^2 \sum_{j=n+1}^{m} \lambda_j.
$$

### A.4. Bounding (II)

For any $v \sim \mu$, we have

$$\mathop{\mathbb{E}}_{v \sim \mu} \left[ \left\| \left( \widehat{\mathcal{F}}_n - \mathcal{F} \right) \left( v - \Pi_m(v) \right) \right\|_{L^2}^2 \right] \leq \left\| \widehat{\mathcal{F}}_n - \mathcal{F} \right\|_{\mathrm{op}}^2 \mathbb{E} \left[ \| v - \Pi_m(v) \|_{L^2}^2 \right]$$

Let $v = \sum_{j \geq 1} \sqrt{\lambda_j} \, \xi_j \varphi_j$. Then,

$$\mathbb{E} \left[ \| v - \Pi_m(v) \|_{L^2}^2 \right] = \mathbb{E} \left[ \left\| v - \sum_{j=1}^m \sqrt{\lambda_j} \, \xi_j \, \varphi_j \right\|_{L^2}^2 \right]$$

$$= \mathbb{E} \left[ \int_{\mathcal{X}} \left( v(x) - \sum_{j=1}^m \sqrt{\lambda_j} \, \xi_j \, \varphi_j(x) \right)^2 d\nu(x) \right]$$

$$= \int_{\mathcal{X}} \mathbb{E} \left[ \left( v(x) - \sum_{j=1}^m \sqrt{\lambda_j} \, \xi_j \, \varphi_j(x) \right)^2 \right] d\nu(x)$$

$$\leq \nu(\mathcal{X}) \cdot \sup_{x \in \mathcal{X}} \mathbb{E} \left[ \left( v(x) - \sum_{j=1}^m \sqrt{\lambda_j} \, \xi_j \, \varphi_j(x) \right)^2 \right].$$

The third equality uses joint measurability, finiteness of $\nu$, and Tonelli's theorem to exchange the order or integration. Therefore, we have established that

$$(\text{II}) \leq \left\| \widehat{\mathcal{F}}_n - \mathcal{F} \right\|_{\mathrm{op}}^2 \cdot \nu(\mathcal{X}) \cdot \sup_{x \in \mathcal{X}} \mathbb{E} \left[ \left( v(x) - \sum_{j=1}^m \sqrt{\lambda_j} \, \xi_j \, \varphi_j(x) \right)^2 \right].$$

### A.5. Combining (I) and (II)

For each $m > n$, we have established

$$\mathop{\mathbb{E}}_{v \sim \mu} \left[ \left\| \widehat{\mathcal{F}}_n(v) - \mathcal{F}(v) \right\|_{L^2}^2 \right] \leq (\text{I}) + 2\sqrt{(\text{I}) \, (\text{II})} + (\text{II}),$$

where

$$(\text{I}) \leq \varepsilon^2 \sum_{i=1}^n \lambda_i + \| \mathcal{F} \|_{\mathrm{op}}^2 \sum_{j=n+1}^m \lambda_j$$

$$(\text{II}) \leq \left\| \widehat{\mathcal{F}}_n - \mathcal{F} \right\|_{\mathrm{op}}^2 \cdot \nu(\mathcal{X}) \cdot \sup_{x \in \mathcal{X}} \mathbb{E} \left[ \left( v(x) - \sum_{j=1}^m \sqrt{\lambda_j} \, \xi_j \, \varphi_j(x) \right)^2 \right].$$

It now remains to choose $m$ such that the upperbound is minimized. To that end, we will take $m \to \infty$. Since $w_1, \ldots, w_n \in L^2(\mathcal{X})$, we must have $\left\| \widehat{\mathcal{F}}_n \right\|_{\mathrm{op}} < \infty$. As $\mathcal{F}$ is also a bounded linear operator, for any $n \in \mathbb{N}$, we must have

$$\left\| \widehat{\mathcal{F}}_n - \mathcal{F} \right\|_{\mathrm{op}}^2 < \infty.$$

Importantly, the norm of $\widehat{\mathcal{F}}_n - \mathcal{F}$ may grow with $n$, but is independent of $m$ and does not grow as $m \to \infty$. Moreover, as $\nu$ is a finite measure, we must have $\nu(\mathcal{X}) < \infty$. Therefore, Karhunen–Loève Theorem (Hsing & Eubank, 2015, Theorem

7.3.5) (also see Equation (2)) implies that

$$\text{(II)} \le \left\| \widehat{\mathcal{F}}_n - \mathcal{F} \right\|_{\text{op}}^2 \cdot \nu(\mathcal{X}) \cdot \sup_{x \in \mathcal{X}} \mathbb{E}\left[ \left( v(x) - \sum_{j=1}^{m} \sqrt{\lambda_j}\, \xi_j\, \varphi_j(x) \right)^2 \right] \xrightarrow{m \to \infty} 0.$$

On the other hand,

$$\text{(I)} \xrightarrow{m \to \infty} \varepsilon^2 \sum_{i=1}^{n} \lambda_i + \|\mathcal{F}\|_{\text{op}}^2 \sum_{j=n+1}^{\infty} \lambda_j.$$

Overall, we have shown that

$$\mathbb{E}_{v \sim \mu}\left[ \left\| \widehat{\mathcal{F}}_n(v) - \mathcal{F}(v) \right\|_{L^2}^2 \right] \le \varepsilon^2 \sum_{i=1}^{n} \lambda_i + \|\mathcal{F}\|_{\text{op}}^2 \sum_{j=n+1}^{\infty} \lambda_j.$$

This completes our proof of Theorem 3.1.

## B. Examples of Covariance Kernels

In this section, we build upon and present a more rigorous analysis of the material discussed in Section 3.3 of the main text.

### B.1. Fractional Inverse of Shifted Laplacian

Li et al. (2021) and Kovachki et al. (2023) generated input functions from $\text{GP}(0, \alpha(-\nabla^2 + \beta \mathbf{I})^{-\gamma})$ for some constants $\alpha, \beta, \gamma > 0$. Here, $\nabla^2$ is the Laplacian operator defined as

$$\nabla^2 v = \sum_{j=1}^{d} \frac{\partial^2 v}{\partial x_j^2}.$$

In this section, we will consider $\mathcal{X}$ to be a $d$-dimensional periodic torus $\mathbb{T}^d$ and the base measure $\nu$ is Lebesgue. We identify $\mathbb{T}^d$ by $[0,1]^d$ with periodic boundary conditions.

Let us define a function $\varphi_m : \mathbb{T}^d \to \mathbb{C}$ as $\varphi_m(x) = e^{2\pi \, \mathrm{i}\, m \cdot x}$ for every $m \in \mathbb{Z}^d$. Recall that $\varphi_m$ is the eigenfunction of $\nabla^2$ with eigenvalue $-4\pi^2 |m|_2^2$. In particular,

$$\nabla^2 e^{2\pi \, \mathrm{i}\, m \cdot x} = \sum_{j=1}^{d} \frac{\partial^2}{\partial x_j^2} e^{2\pi \, \mathrm{i}\, m \cdot x} = \sum_{j=1}^{d} (2\pi \, \mathrm{i}\, m_j)^2 e^{2\pi \, \mathrm{i}\, m \cdot x} = -4\pi^2 |m|_2^2\, e^{2\pi \, \mathrm{i}\, m \cdot x}.$$

Since $\{\varphi_m \, : \, m \in \mathbb{Z}^d\}$ forms a complete orthonormal system in $L^2(\mathbb{T}^d)$, there are no other eigenfunctions of $\nabla^2$. A simple algebra shows that $\varphi_m$'s are also he eigenfunctions of shifted Laplacian $-\nabla^2 + \beta \mathbf{I}$ with eigenvalues being $(\beta + 4\pi^2 |m|_2^2)$. Finally, the spectral mapping theorem implies that $\{(\lambda_m, \varphi_m) \, : \, m \in \mathbb{Z}^d\}$ is the sequence of eigenpairs of $\alpha(-\nabla^2 + \beta \mathbf{I})^{-\gamma}$, where the eigenvalues are

$$\lambda_m = \alpha \left( \beta + 4\pi^2 |m|_2^2 \right)^{-\gamma}.$$

Next, we need to show that these eigenvalues are summable to use Theorem 3.1. Note that

$$\sum_{m \in \mathbb{Z}^d} \lambda_m = \sum_{m \in \mathbb{Z}^d} \alpha \left( \beta + 4\pi^2 |m|_2^2 \right)^{-\gamma} \le \alpha \beta^{-\gamma} + \frac{\alpha}{(2\pi)^{2\gamma}} \sum_{m \in \mathbb{Z}^d \setminus \{0\}} |m|_\infty^{-2\gamma}.$$

It is easy to see that

$$\sum_{m \in \mathbb{Z}^d \setminus \{0\}} |m|_\infty^{-2\gamma} \le \sum_{j=1}^{\infty} j^{-2\gamma} (2j+1)^d \le 3^d \sum_{j=1}^{\infty} j^{-2\gamma+d} < \infty$$

as long as $2\gamma > d$. The first inequality holds because $|\{m \in \mathbb{Z}^d \setminus \{0\} \, : \, |m|_\infty = j\}| \le 2(2j+1)^{d-1}$. This is true because at least one of the entries has to be $\pm j$ and other $d-1$ entries could be anything in $\{0, \pm 1, \ldots, \pm j\}$. So we have $\sum_{m \in \mathbb{Z}^d} |\lambda_m| < \infty$, implying that the operator $\alpha(-\nabla^2 + \beta \mathbf{I})^{-\gamma}$ is in trace class as long as $2\gamma > d$.

Finally, it is easy to see that the operator $\alpha(-\nabla^2 + \beta\mathbf{I})^{-\gamma}$ is integral operator associated with the kernel

$$K(y,x) = \sum_{m \in \mathbb{Z}^d} \lambda_m \, \varphi_m(y) \, \overline{\varphi_m(x)}.$$

Upon writing the Fourier series of $v \in L^2(\mathbb{T})$, it is obvious that $\left(\alpha(-\nabla^2 + \beta\mathbf{I})^{-\gamma}v\right)(y) = \int_{\mathcal{X}} K(y,x)v(x)\,dx$ for all $y \in \mathcal{X}$. As $\sum_{m \in \mathbb{Z}} |\lambda_m| < \infty$, the convergence is absolute and uniform. Since $K$ is a uniform limit of the sum of continuous functions, $K$ must also be continuous. Moreover, as $\lambda_m = \lambda_{-m}$, the kernel $K$ must be real-valued. In particular, we have

$$\begin{aligned}
K(y,x) &= \sum_{m \in \mathbb{Z}^d} \lambda_m \, \varphi_m(y) \, \overline{\varphi_m(x)} \\
&= \sum_{m \in \mathbb{Z}^d} \frac{\lambda_m}{2} \left( \varphi_m(y) \, \overline{\varphi_m(x)} + \varphi_{-m}(y) \, \overline{\varphi_{-m}(x)} \right) \\
&= \sum_{m \in \mathbb{Z}^d} \lambda_m \cos\left( 2\pi m \cdot (y - x) \right) \\
&= \sum_{m \in \mathbb{Z}^d} \lambda_m \left( \cos(2\pi m \cdot y) \cos(2\pi m \cdot x) + \sin(2\pi m \cdot y) \sin(2\pi m \cdot x) \right)
\end{aligned}$$

This is a generalization of the cosine covariance kernel often considered in computational PDE literature (see (Lord et al., 2014, Example 5.20)). Since $\cos(\theta) = \cos(-\theta)$, it is obvious that $K$ is symmetric. As $K$ is a continuous and real-valued covariance kernel defined on a bounded domain $\mathbb{T}^d$, we can use Theorem 3.1 for such $K$. In principle, we could use the Fourier modes $\varphi_m$'s as source terms to define the estimator discussed in Section 3.1. However, the proof of Theorem 3.1 assumes that the eigenfunctions of the kernel are real-valued. So, we will first show that we can write the eigenfunctions of $K$ solely using sine and cosine functions without having to use complex exponentials. This allows us to use these sine and cosine functions to define the estimator discussed in Section 3.1, and invoke results of Theorem 3.1.

### B.1.1. CASE $d = 1$

When $d = 1$, it is easy to see that $\{1\} \cup \{\sqrt{2}\cos(2\pi jx), \sqrt{2}\sin(2\pi jx) : j \in \mathbb{N}\}$ are the eigenfunctions of $K$. Writing the expansion of $K$ and using Fubini's to switch the sum and the integral, we get

$$\int_{\mathbb{T}} K(y,x)\sqrt{2}\cos(2\pi jx)\,dx = \lambda_j \sqrt{2} \, \cos(2\pi jy)\frac{1}{2} + \lambda_{-j}\sqrt{2} \, \cos(-2\pi jy)\frac{1}{2} = \lambda_j \sqrt{2}\cos(2\pi jy).$$

Note that the first equality holds because $\cos(2\pi jx)$ is orthogonal to all other cosine and sine functions except for $\cos(2\pi jx)$ and $\cos(-2\pi jx)$. The final equality holds because $\lambda_j = \lambda_{-j}$ and $\cos(\theta) = \cos(-\theta)$. A similar calculation shows that

$$\int_{\mathbb{T}} K(y,x)\sqrt{2}\sin(2\pi jx)\,dx = \lambda_j \sqrt{2} \, \sin(2\pi jy)\frac{1}{2} + \lambda_{-j}\sqrt{2} \, \sin(-2\pi jy)\frac{-1}{2} = \lambda_j \sqrt{2}\sin(2\pi jy).$$

Finally, we have $\int_{\mathbb{T}} K(y,x)\,1\,dx = \lambda_0 1$. Thus, $\lambda_j$ for $j \in \mathbb{N}$ are the eigenvalues for sine/cosine functions and $\lambda_0$ for 1. Since $\{1\} \cup \{\sqrt{2}\cos(2\pi jx), \sqrt{2}\sin(2\pi jx) : j \in \mathbb{N}\}$ forms a complete orthonormal system of $L^2(\mathbb{T}, \mathbb{R})$, there cannot be any more eigenfunctions of $K$. Next, we will plug in the values of $\lambda_j$'s in Theorem 3.1 to get the precise rates.

Pick an odd $n \in \mathbb{N}$ and suppose the $n$ input terms used to construct the estimator in Section 3.1 are $\{1\} \cup$

$\{\sqrt{2}\cos(2\pi jx), \sqrt{2}\sin(2\pi jx) : j \leq (n-1)/2\}$. Then, the upperbound is

$$\varepsilon^2 \left( \lambda_0 + \sum_{j=1}^{(n-1)/2} 2\lambda_j \right) + \|\mathcal{F}\|_{\text{op}}^2 \sum_{j=(n+1)/2}^{\infty} 2\lambda_j$$

$$\leq \varepsilon^2 \alpha \beta^{-\gamma} + \varepsilon^2 2 \sum_{j=1}^{(n-1)/2} \alpha \left( \beta + 4\pi^2 j^2 \right)^{-\gamma} + 2\|\mathcal{F}\|_{\text{op}}^2 \sum_{j=(n+1)/2}^{\infty} \alpha \left( \beta + 4\pi^2 j^2 \right)^{-\gamma}$$

$$\leq \varepsilon^2 \alpha \beta^{-\gamma} + \varepsilon^2 \alpha \frac{2}{(4\pi^2)^\gamma} \sum_{j=1}^{(n-1)/2} \frac{1}{j^{2\gamma}} + \frac{2\alpha}{(4\pi^2)^\gamma} \|\mathcal{F}\|_{\text{op}}^2 \sum_{j=(n+1)/2}^{\infty} \frac{1}{j^{2\gamma}}$$

$$\leq \varepsilon^2 \alpha \beta^{-\gamma} + \varepsilon^2 \alpha \frac{2}{(4\pi^2)^\gamma} + \varepsilon^2 \alpha \frac{2}{(4\pi^2)^\gamma} \int_1^{(n-1)/2} t^{-2\gamma}\, dt + \frac{2\alpha}{(4\pi^2)^\gamma} \|\mathcal{F}\|_{\text{op}}^2 \int_{(n-1)/2}^{\infty} t^{-2\gamma}\, dt$$

$$\leq \varepsilon^2 \alpha \beta^{-\gamma} + \varepsilon^2 \alpha \frac{2}{(4\pi^2)^\gamma} + \frac{2}{(4\pi^2)^\gamma} \frac{\varepsilon^2 \alpha}{2\gamma - 1} + \frac{2\alpha}{(4\pi^2)^\gamma} \|\mathcal{F}\|_{\text{op}}^2 \frac{1}{2\gamma - 1} \frac{2^{2\gamma - 1}}{(n-1)^{2\gamma - 1}}, \quad \forall \gamma > \frac{1}{2}.$$

Since $2 \cdot 2^{2\gamma - 1} \leq (4\pi^2)^\gamma$ and $2^{2\gamma - 1} \leq (4\pi^2)^\gamma$, the overall error is at most

$$\varepsilon^2 \left( \alpha \beta^{-\gamma} + \alpha + \frac{\alpha}{2\gamma - 1} \right) + \frac{\alpha \|\mathcal{F}\|_{\text{op}}^2}{2\gamma - 1} \frac{1}{(n-1)^{2\gamma - 1}} \qquad \text{for all } \gamma > \frac{1}{2}.$$

Since $\gamma > 1/2$, the reducible error goes to 0 as $n \to \infty$. As an example, (Li et al., 2021) uses $\alpha = 625$, $\beta = 25$ and $\gamma = 2$ in their experiment for $1d$-Burger's equation. In this case, we get the convergence rate of $n^{-3}$ for the reducible error. Note that this rate of *cubic order* is faster than the usual passive statistical rate of $1/n$. In fact, for any value $\tau$, one can take $\gamma = (\tau + 1)/2$ to get the rate of $n^{-\tau}$. Thus, every polynomial rate is possible depending on the choice of $\gamma$.

### B.1.2. CASE $d > 1$

Recall that $\{1\} \cup \{\sqrt{2}\cos(2\pi jx), \sqrt{2}\sin(2\pi jx) : j \in \mathbb{N}\}$ are the eigenvalues of $K$ for $d = 1$ with eigenvalues $\lambda_j := \alpha \left( \beta + 4\pi^2 j^2 \right)^{-\gamma}$. Define a set of functions

$$\mathcal{E} = \prod_{i=1}^{d} \{1\} \cup \{\sqrt{2}\cos(2\pi jx_i), \sqrt{2}\sin(2\pi jx_i) : j \in \mathbb{N}\}.$$

For each element $e \in \mathcal{E}$, there exists a tuple $j := (j_1, \ldots, j_d) \in \mathbb{N}_0^d$ such that

$$e(x) = \psi_{j_1}(x_1) \ldots \psi_{j_{d-1}}(x_{d-1}) \cdot \psi_{j_d}(x_d),$$

where $\psi_{j_i}(x_i) \in \{\sqrt{2}\cos(2\pi j_i x_i), \sqrt{2}\sin(2\pi j_i x_i)\}$ for $j_i > 0$ and $\sqrt{2}$ for $j_i = 0$. Let us denote the collection of all such functions by $\mathcal{E}_j$. Then, we have $\mathcal{E} = \cup_{j \in \mathbb{N}_0^d} \mathcal{E}_j$. We prove the following result on the eigenpairs of $K$.

**Proposition B.1.** *For each $j \in \mathbb{N}_0^d$, define $\lambda_j = \alpha \left( \beta + 4\pi^2 |j|_2^2 \right)^{-\gamma}$. Then,*

$$\bigcup_{j \in \mathbb{N}_0^d} \bigcup_{e \in \mathcal{E}_j} \{(\lambda_j, e)\}$$

*is the set of eigenpairs of $K$ on $\mathbb{T}^d$.*

We defer the full proof of Proposition B.1 to the end of this subsection. First, we use Proposition B.1 and Theorem 3.1 to get the precise rate for kernel $K$. Pick $r$ such that the source terms used to construct the estimator defined in Section 3.1 are

$$\bigcup_{j \in \mathbb{N}_0^d : |j|_\infty \leq r} \mathcal{E}_j$$

Note that $|\mathcal{E}_0| = 1$ and $|\mathcal{E}_j| \leq 2^d$ for all $|j|_\infty > 0$. Thus, there are $n \leq (r+1)^d 2^d$ source terms. Then, the upperbound is

$$\leq \varepsilon^2 \lambda_0 + \sum_{0 < |j|_\infty \leq r} 2^d \lambda_j + \|\mathcal{F}\|_{\text{op}}^2 \sum_{|j|_\infty > r} 2^d \lambda_j$$

$$= \varepsilon^2 \alpha \beta^{-\gamma} + \varepsilon^2 2^d \sum_{0 < |j|_\infty \leq r} \alpha \left(\beta + 4\pi^2 |j|_2^2\right)^{-\gamma} + 2^d \|\mathcal{F}\|_{\text{op}}^2 \sum_{|j|_\infty > r} \alpha \left(\beta + 4\pi^2 |j|_2^2\right)^{-\gamma}$$

$$= \varepsilon^2 \alpha \beta^{-\gamma} + \varepsilon^2 2^d \sum_{0 < |j|_\infty \leq r} \alpha \left(\beta + 4\pi^2 |j|_\infty^2\right)^{-\gamma} + 2^d \|\mathcal{F}\|_{\text{op}}^2 \sum_{|j|_\infty > r} \alpha \left(\beta + 4\pi^2 |j|_\infty^2\right)^{-\gamma}$$

$$\leq \varepsilon^2 \alpha \beta^{-\gamma} + \varepsilon^2 2^d \sum_{k=1}^{r} \alpha \left(\beta + 4\pi^2 k^2\right)^{-\gamma} (k+1)^{d-1} + 2^d \|\mathcal{F}\|_{\text{op}}^2 \sum_{k=r+1}^{\infty} \alpha \left(\beta + 4\pi^2 k^2\right)^{-\gamma} (k+1)^{d-1}$$

$$\leq \varepsilon^2 \alpha \beta^{-\gamma} + \varepsilon^2 2^d \frac{\alpha 2^d}{(4\pi^2)^\gamma} \sum_{k=1}^{r} k^{d-1-2\gamma} + 2^d \|\mathcal{F}\|_{\text{op}}^2 \frac{\alpha 2^d}{(4\pi^2)^\gamma} \sum_{k=r+1}^{\infty} k^{d-1-2\gamma}$$

$$\leq \varepsilon^2 \alpha \beta^{-\gamma} + \varepsilon^2 \frac{\alpha 2^{2d}}{(4\pi^2)^\gamma} + \varepsilon^2 \frac{\alpha 2^{2d}}{(4\pi^2)^\gamma} \int_1^r t^{d-1-2\gamma} \, dt + \|\mathcal{F}\|_{\text{op}}^2 \frac{\alpha 2^{2d}}{(4\pi^2)^\gamma} \int_r^\infty t^{d-1-2\gamma} \, dt$$

$$\leq \varepsilon^2 \alpha \beta^{-\gamma} + \varepsilon^2 \frac{\alpha 2^{2d}}{(4\pi^2)^\gamma} + \varepsilon^2 \frac{\alpha 2^{2d}}{(4\pi^2)^\gamma} \frac{1}{2\gamma - d} + \|\mathcal{F}\|_{\text{op}}^2 \frac{\alpha 2^{2d}}{(4\pi^2)^\gamma} \frac{1}{2\gamma - d} \frac{1}{r^{2\gamma - d}}, \qquad \text{for all } 2\gamma > d.$$

Recall that $n \leq (2r+2)^d$. So, we have $n^{1/d}/2 - 1 \leq r$. For $n^{1/d} \geq 4$, we have $r \geq n^{1/d}/4$. Thus,

$$\frac{1}{r^{2\gamma - d}} \leq \frac{4^{2\gamma - d}}{n^{\frac{2\gamma}{d} - 1}}$$

Note that $(4\pi^2)^\gamma = (2\pi)^{2\gamma} \geq 2^{2d} 4^{2\gamma - d}$. Moreover, as $2\gamma > d$, we also have $(4\pi^2)^\gamma \geq 2^{2d}$. Therefore, our upper bound is at most

$$\varepsilon^2 \left(\alpha \beta^{-\gamma} + \alpha + \frac{\alpha}{2\gamma - d}\right) + \frac{\alpha \|\mathcal{F}\|_{\text{op}}^2}{2\gamma - d} \frac{1}{n^{\frac{2\gamma}{d} - 1}}.$$

Since $2\gamma/d - 1 > 0$, the reducible error above goes to 0 when $n \to \infty$. Again, as an example, (Li et al., 2021) uses $\alpha = 7^{3/2}$, $\beta = 49$ and $\gamma = 2.5$ in their experiment for $2d$-Navier Stokes. In this case, $2\gamma/d = 2.5$, yielding the convergence rate of $n^{-1.5}$ for the reducible error. Note that this rate is faster than the usual passive statistical rate of $1/n$. However, as usual, for any value $\tau$, one can take $\gamma = d(\tau + 1)/2$ to get the rate of $n^{-\tau}$. Thus, every polynomial rate is possible depending on the choice of $\gamma$.

We now end this section by providing the proof of Theorem B.1.

*Proof of Proposition B.1.* Since $\cup_{j \in \mathbb{N}_0^d} \cup_{e \in \mathcal{E}_j} \{e\}$ forms an orthonormal basis of $L^2(\mathbb{T}^d, \mathbb{R})$, there cannot be anymore eigenfunctions of $K$. Thus, it suffices to show that $(\lambda_j, e)$ is an eigenpair for any $e \in \mathcal{E}_j$ and $j \in \mathbb{N}_0^d$. To prove this, we will establish that

$$\int_{\mathbb{T}^d} \sum_{m \in \mathbb{Z}^d} \mathbb{1}\{|m_i| = j_i \; \forall i \in [d]\} \cos\left(2\pi m \cdot (y - x)\right) e_j(x) \, dx = e_j(y), \tag{3}$$

where $e_j$ is an arbitrary element of $\mathcal{E}_j$. Recall that

$$\int_{\mathbb{T}^d} \cos\left(2\pi m \cdot (y - x)\right) e_j(x) \, dx = 0 \text{ if } \exists i \text{ such that } |m_i| \neq j_i.$$

This is true because if $\exists i$ such that $|m_i| \neq j_i$, then we can write $\cos\left(2\pi m \cdot (y - x)\right) = \cos\left(2\pi \sum_{\ell \neq i} m_\ell(y_\ell - x_\ell)\right) \cos(2\pi m_i(y_i - x_i)) - \sin\left(2\pi \sum_{\ell \neq i} m_\ell(y_\ell - x_\ell)\right) \sin(2\pi m_i(y_i - x_i))$. Moreover, $e_j(x) = \psi_{j_1}(x_1) \ldots \psi_{j_{d-1}}(x_{d-1}) \cdot \psi_{j_d}(x_d)$, where $\psi_{j_\ell}$'s are either sine, cosine, or a constant function. Our claim follows upon noting that $\psi_{j_i}(x_i)$ is orthogonal to both $\sin(2\pi m_i(y_i - x_i))$ and $\cos(2\pi m_i(y_i - x_i))$.

Thus, Equation (3) together with the fact that $\lambda_m = \lambda_j$ for all $m \in \{k \in \mathbb{Z}^d : |k_i| = j_i \; \forall i \in [d]\}$ implies that $(\lambda_j, e_j)$ is the eigenpair of $K$. As $j \in \mathbb{N}_0^d$ and $e_j \in \mathcal{E}_j$ are arbitrary, this completes our proof.

Now, it remains to prove Equation (3). We will proceed by induction on $d$. For the base case, take $d = 1$. If $j = 0$, $e_j = 1$ and our claim follows trivially. Suppose $j \neq 0$. Since $\cos(\theta) = \cos(-\theta)$, we have

$$\sum_{m \in \mathbb{Z}} \mathbb{1}\{|m| = j\} \cos\left(2\pi m(y - x)\right) = 2\cos(2\pi j(y - x))$$

$$= 2\cos(2\pi jy)\cos(2\pi jx) + 2\sin(2\pi jy)\sin(2\pi jx).$$

If $e_j(x) = \sqrt{2}\cos(2\pi jx)$, then

$$\int_{\mathbb{T}} \left(2\cos(2\pi jy)\cos(2\pi jx) + 2\sin(2\pi jy)\sin(2\pi jx)\right) \sqrt{2}\cos(2\pi jx)\, dx = \sqrt{2}\cos(2\pi jy).$$

If $e_j(x) = \sqrt{2}\sin(2\pi jx)$, a similar calculation shows that

$$\int_{\mathbb{T}} \left(2\cos(2\pi jy)\cos(2\pi jx) + 2\sin(2\pi jy)\sin(2\pi jx)\right) \sqrt{2}\sin(2\pi jx)\, dx = \sqrt{2}\sin(2\pi jy).$$

This completes our proof of the base case.

Suppose (3) is true for $d - 1$. We will now prove it for $d$. Note that

$$\cos\left(2\pi m \cdot (y - x)\right) = \cos\left(2\pi \sum_{i=1}^{d} m_i(y_i - x_i)\right)$$

$$= \cos\left(2\pi \sum_{i=1}^{d-1} m_i(y_i - x_i)\right) \cos\left(2\pi m_d(y_d - x_d)\right) - \sin\left(2\pi \sum_{i=1}^{d-1} m_i(y_i - x_i)\right) \sin\left(2\pi m_d(y_d - x_d)\right).$$

First, observe that when summed over all $m \in \mathbb{Z}^d$ such that $|m_i| = j_i$ for all $i \in [d]$, the sine term vanishes. That is,

$$\sum_{m \in \mathbb{Z}^d} \mathbb{1}\{|m_i| = j_i \ \forall i \in [d]\} \left(\sin\left(2\pi \sum_{i=1}^{d-1} m_i(y_i - x_i)\right) \sin\left(2\pi m_d(y_d - x_d)\right)\right)$$

$$= \left(\sum_{m \in \mathbb{Z}^{d-1}} \mathbb{1}\{|m_i| = j_i \ \forall i \in [d-1]\} \sin\left(2\pi \sum_{i=1}^{d-1} m_i(y_i - x_i)\right)\right) \left(\sum_{m_d \in \mathbb{Z}} \mathbb{1}\{|m_d| = j_d\} \sin\left(2\pi m_d(y_d - x_d)\right)\right)$$

$$= 0.$$

The final step follows here because the term in the second parenthesis above is always $0$. There are two cases to consider. If $j_d = 0$, the summand only has one term and our claim holds as $\sin(0) = 0$. On the other hand, if $j_d \neq 0$, then we are have $\sin(\theta) + \sin(-\theta) = 0$.

Therefore, we obtain

$$\sum_{m \in \mathbb{Z}^d} \mathbb{1}\{|m_i| = j_i \ \forall i \in [d]\} \cos\left(2\pi m \cdot (y - x)\right)$$

$$= \left(\sum_{m \in \mathbb{Z}^{d-1}} \mathbb{1}\{|m_i| = j_i \ \forall i \in [d-1]\} \cos\left(2\pi \sum_{i=1}^{d-1} m_i(y_i - x_i)\right)\right) \left(\sum_{m_d \in \mathbb{Z}} \mathbb{1}\{|m_d| = j_d\} \cos\left(2\pi m_d(y_d - x_d)\right)\right)$$

A similar factorization can be done for $e_j$ to write

$$e_j(x) = \psi_{j_1}(x_1) \dots \psi_{j_d}(x_d),$$

where $\psi_{j_i}$'s are either sine, cosine, or a constant function.

However, $\psi_{j_d}$ is some $e_{j_d}$ defined on $\mathbb{T}$. Thus, using the base case, we have

$$\int_{\mathbb{T}} \sum_{m_d \in \mathbb{Z}} \mathbb{1}\{|m_d| = j_d\} \cos\left(2\pi m_d(y_d - x_d)\right) \psi_{j_d}(x_d)\, dx_d = \psi_{j_d}(y_d).$$

Similarly, using the induction hypothesis, we have

$$\int_{\mathbb{T}^{d-1}} \left( \sum_{m \in \mathbb{Z}^{d-1}} \mathbb{1}\{|m_i| = j_i \ \forall i \in [d-1]\} \cos\left(2\pi \sum_{i=1}^{d-1} m_i(y_i - x_i)\right)\right) \prod_{i=1}^{d-1} \psi_{j_i}(x_i) \ d(x_1, \ldots, x_{d-1})$$
$$= \prod_{i=1}^{d-1} \psi_{j_i}(y_i).$$

Combining everything, we obtain

$$\int_{\mathbb{T}^d} \sum_{m \in \mathbb{Z}^d} \mathbb{1}\{|m_i| = j_i \ \forall i \in [d]\} \cos\left(2\pi m \cdot (y - x)\right) \prod_{i=1}^{d} \psi_{j_i}(x_i) \ dx = \prod_{i=1}^{d} \psi_{j_i}(y_i).$$

The final step requires using the factorization of cosine and writing integral over $\mathbb{T}^d$ as product of integral over $\mathbb{T}^{d-1}$ and $\mathbb{T}$. This completes our induction step, and thus the proof. $\qquad\square$

### B.2. RBF Kernel on $\mathbb{R}$.

Let $K$ be the RBF kernel on $\mathbb{R}$. That is, $K(x,y) = \exp\left(-\frac{1}{2\ell^2}|x-y|^2\right)$ for all $x, y \in \mathbb{R}$. For now, let $\nu$ is a Gaussian measure with mean 0 and variance $\sigma^2$ on $\mathbb{R}$. Then, it is known (Williams & Rasmussen, 2006, Section 4.3.1) that $K(x,y) = \sum_{j=0}^{\infty} \lambda_j \varphi_j(x) \varphi_j(y)$, where

$$\lambda_j := \sqrt{\frac{2a}{a+b+c}} \left(\frac{b}{a+b+c}\right)^j$$
$$\varphi_j(x) := \exp(-(c-a)x^2) H_j(\sqrt{2c}x).$$

Here, $a = (4\sigma^2)^{-1}$, $b = (2\ell^2)^{-1}$, $c = \sqrt{a^2 + 2ab}$, and $H_j(\cdot)$ is the Hermite polynomial of order $j$ defined as

$$H_j(x) = (-1)^j \exp(x^2) \frac{d^j}{dx^j} \exp(-x^2).$$

Note that this is the eigenpairs of $K(y,x)$ over the entire $\mathbb{R}$, whereas we need eigenpairs over some compact domain $\mathcal{X} \subseteq \mathbb{R}$. The eigenpairs of $K(y,x)$ are generally not available in closed form for arbitrary $\mathcal{X}$. However, the variance of the Gaussian measure $\sigma^2$ can be tuned appropriately to localize the domain $\mathbb{R}$ to appropriate $\mathcal{X}$ of interest. For example, let $\mathcal{X} = [-1, 1]$. Then,

$$\int_{-1}^{1} K(y,x)\varphi_j(x) \, d\nu(x) = \int_{\mathbb{R}} K(y,x) \varphi_j(x) \, d\nu(x) - \int_{|x|>1} K(y,x) \varphi_j(x) \, d\nu(x).$$

Since $\int_{\mathbb{R}} K(y,x) \varphi_j(x) \, d\nu(x) = \lambda_j \varphi_j(y)$, we have

$$\left|\int_{-1}^{1} K(y,x)\varphi_j(x) \, d\nu(x) - \lambda_j \varphi_j(y)\right| \leq \int_{|x|>1} |K(y,x)| \, |\varphi_j(x)| \, d\nu(x)$$
$$\leq \sqrt{\int_{|x|>1} |K(y,x)|^2 \, d\nu(x)} \sqrt{\int_{|x|>1} |\varphi_j(x)|^2 \, d\nu(x)}$$
$$\leq \sqrt{\int_{|x|>1} \exp\left(-\frac{|x-y|^2}{\ell^2}\right) \, d\nu(x)},$$

where the second term is upper bounded by 1 as $\varphi_j^2$ integrates to 1 over the whole domain $\mathbb{R}$. Note that $\exp\left(-\frac{|x-y|^2}{\ell^2}\right) \leq 1$ and $\sqrt{\nu([-1,1]^c)} \leq 3.9 \times 10^{-12}$ when $\sigma = 0.1$. So, $\sigma$ can be appropriately tuned such that $(\lambda_j, \varphi_j)_{j \geq 1}$ is a good approximation of the eigenpair of $K$ for our domain $\mathcal{X}$ of interest. Next, we use these eigenvalues to study how the upper bound in Theorem 3.1 decays as $n \to \infty$.

Let $\gamma := b/(a + b + c)$. It is clear that $\gamma \in (0, 1)$. Since $c = \sqrt{a^2 + 2ab} \geq a$, we also have $\sqrt{\frac{2a}{a+b+c}} \leq 1$. Thus, we obtain $\lambda_j \leq \gamma^j$. Plugging this estimate in the upperbound of Theorem 3.1, we obtain

$$\varepsilon^2 \sum_{i=0}^{n-1} \lambda_i + \|\mathcal{F}\|_{\mathrm{op}}^2 \sum_{i=n}^{\infty} \lambda_i \leq \varepsilon^2 \sum_{i=0}^{n-1} \gamma^j + \|\mathcal{F}\|_{\mathrm{op}}^2 \sum_{i=n}^{\infty} \gamma^j$$

$$= \frac{1 - \gamma^n}{1 - \gamma} \varepsilon^2 + \|\mathcal{F}\|_{\mathrm{op}}^2 \frac{\gamma^n}{1 - \gamma}$$

$$\leq \frac{1}{(1 - \gamma)} \left( \varepsilon^2 + \|\mathcal{F}\|_{\mathrm{op}}^2 \gamma^n \right).$$

Therefore, the reducible error vanishes exponentially fast as $n \to \infty$.

### B.3. RBF Kernel on $\mathbb{R}^d$

Let $K(y, x) = \exp(-|x - y|_2^2/(2\ell^2))$, where $x, y \in \mathbb{R}^d$. Then, it is clear that

$$K(y, x) = \prod_{i=1}^{d} \exp(-|x_i - y_i|^2/(2\ell^2)) =: \prod_{i=1}^{d} K_i(y_i, x_i).$$

If $(\lambda_{ij}, \varphi_{ij})_{j \in \mathbb{N}}$ are the eigenpairs of $K_i$ under the weighted measure standard Gaussian measure on $\mathbb{R}$, then

$$\left\{ \left( \prod_{i=1}^{d} \lambda_{ij_i} , \prod_{i=1}^{d} \varphi_{ij_i} \right) \middle| (j_1, j_2, \ldots, j_d) \in \mathbb{N}_0^d \right\}$$

are the eigenpairs of $K$ when $\nu$ is multivariate Gaussian with mean $0$ and covariance $\sigma^2 \mathbf{I}$. This follows immediately upon noting that

$$\int_{\mathbb{R}^d} K(y, x) \prod_{i=1}^{d} \varphi_{ij_i}(x_i) \, d\nu(x) = \prod_{i=1}^{d} \int_{\mathbb{R}} K_i(y_i, x_i) \varphi_{ij_i}(x_i) \, d\nu(x_i) = \prod_{i=1}^{d} \lambda_{ij_i} \varphi_{ij_i}(x_i).$$

Finally, these are the only eigenpairs because the product functions $\prod_{i=1}^{d} \varphi_{ij_i}$ for all possible $j_1, \ldots, j_d \in \mathbb{N}_0$ form a complete orthonormal system of $L^2(\mathbb{R}^d)$ under the base measure $\nu$.

Pick $m$ such that $m > d$, and suppose the $n$ source terms in Theorem 3.1 are $\{\varphi_{ij} : i \in [d] \text{ and } 0 \leq j \leq m - 1\}$. That is, we have $n = m^d$ source terms. So, the upperbound is

$$\varepsilon^2 \sum_{j_1=0}^{m-1} \cdots \sum_{j_d=0}^{m-1} \prod_{i=1}^{d} \lambda_{ij_i} + \|\mathcal{F}\|_{\mathrm{op}}^2 \sum_{\substack{(j_1,\ldots,j_d) \in \mathbb{N}_0^d \\ \max\{j_1,\ldots,j_d\} \geq m}} \prod_{i=1}^{d} \lambda_{ij_i}$$

The first summation is

$$\sum_{j_1=0}^{m-1} \cdots \sum_{j_d=0}^{m-1} \prod_{i=1}^{d} \lambda_{ij_i} = \prod_{i=1}^{d} \sum_{j_i=0}^{m-1} \lambda_{ij_i} \leq \prod_{i=1}^{d} \sum_{j_i=0}^{m-1} \gamma^{j_i} \leq \left( \frac{1 - \gamma^m}{1 - \gamma} \right)^d \leq \frac{1}{(1 - \gamma)^d}.$$

On the other hand,

$$\sum_{\substack{(j_1,\ldots,j_d) \in \mathbb{N}_0^d \\ \max\{j_1,\ldots,j_d\} \geq m}} \prod_{i=1}^{d} \lambda_{ij_i} \leq \sum_{\substack{(j_1,\ldots,j_d) \in \mathbb{N}_0^d \\ \max\{j_1,\ldots,j_d\} \geq m}} \gamma^{j_1 + \ldots + j_d} \leq \sum_{r=m}^{\infty} r^d \gamma^r \leq \int_{m-1}^{\infty} r^d \gamma^r \, dr.$$

The second inequality follows because the number of tuple $(j_1, \ldots, j_d)$ that sum to $r$ is $\leq r^d$. It is easy to see that the integral converges faster than $1/n^t$ for every $t \geq 1$. To see this, pick $t \geq 1$. Then, there exists $c > 0$ such that $\gamma^r \leq c \, r^{-dt-1-d}$. Note that $c$ may depend on $\gamma, d$, and $t$, but it does not depend on $r$. Thus, we obtain

$$\int_{m-1}^{\infty} r^d \gamma^r \, dr \leq c \int_{m-1}^{\infty} r^{-dt-1} \, dr = \frac{c}{(m-1)^{dt}}.$$

Since $m = n^{1/d}$, this rate is $c'/n^t$ for some $c'$. That is, our overall upper bound is

$$\varepsilon^2 \frac{1}{(1-\gamma)^d} + \|\mathcal{F}\|_{\mathrm{op}}^2 \frac{c'}{n^t}.$$

for some $c'$ for every $t \geq 1$. Therefore, the reducible error vanishes at a rate faster than every polynomial function of $1/n$.

### B.4. Brownian Motion

Let us consider the case where $\mathcal{X} = [0, 1]$, the base measure $\nu$ is Lebegsue, and the stochastic process in Section 2.2 is Brownian motion. Recall that the Brownian motion is a Gaussian process with covariance kernel

$$K(s, t) = \min(s, t) \qquad s, t \in [0, 1].$$

It is well-known (Hsing & Eubank, 2015, Example 4.6.3) that the eigenpairs of $K$ is given by

$$\lambda_j := \frac{1}{\left(j - \frac{1}{2}\right)^2 \pi^2} \qquad \text{and} \qquad \varphi_j(t) := \sqrt{2} \sin\left(\left(j - \frac{1}{2}\right) \pi t\right) \qquad \forall j \in \mathbb{N}.$$

Plugging this in the upperbound of Theorem 3.1 yields the bound

$$\varepsilon^2 \sum_{j=1}^n \frac{1}{\left(j - \frac{1}{2}\right)^2 \pi^2} + \|\mathcal{F}\|_{\mathrm{op}}^2 \sum_{j=n+1}^\infty \frac{1}{\left(j - \frac{1}{2}\right)^2 \pi^2}$$

$$= \varepsilon^2 \frac{\pi^2}{2} \frac{1}{\pi^2} + \|\mathcal{F}\|_{\mathrm{op}}^2 \sum_{j=n+1}^\infty \frac{1}{\left(j - \frac{1}{2}\right)^2 \pi^2}$$

$$\leq \frac{\varepsilon^2}{2} + \|\mathcal{F}\|_{\mathrm{op}}^2 \frac{1}{\pi^2} \int_n^\infty \frac{1}{(t - 1/2)^2} \, dt$$

$$= \frac{\varepsilon^2}{2} + \|\mathcal{F}\|_{\mathrm{op}}^2 \frac{1}{\pi^2} \frac{2}{2n - 1}.$$

Therefore, the reducible error vanishes at rate $\sim \frac{1}{n}$.

## C. Numerical Approximation of Eigenfunctions

In Section 3.3, we provided analytic expressions for the eigenfunctions of certain covariance kernels. However, for some kernels of interest, closed-form expressions for the eigenfunctions are generally not available. In such cases, numerical approximation is necessary. Here, we will briefly mention some key concepts behind the numerical approximation of eigenfunctions of kernels. The material presented here is based on (Williams & Rasmussen, 2006, Section 4.3.2), so we refer the reader to that text for a more detailed discussion and relevant references.

Let $d\nu(x) \propto p(x) \, dx$ for some density function $p$. For example, if $\nu$ is Lebesgue measure on $[-1, 1] \times [-1, 1]$, then $p(x) = 1/4$. Then, the solution of Feldolm integral

$$\int_{\mathcal{X}} K(y, x) \, \varphi_j(x) \, d\nu(x) = \lambda_j \varphi(y)$$

is approximated using the equation

$$\frac{1}{N} \sum_{i=1}^N K(y, x_i) \, \varphi_j(x_i) = \lambda_j \varphi_j(y).$$

Here, $x_1, x_2, \ldots, x_N$ are iid samples from $p$. Taking $y = x_1, \ldots, x_N$, we obtain a matrix eigenvalue equation

$$\mathbf{K} u_j = \gamma_j \, u_j,$$

where $\mathbf{K}$ is a $N \times N$ matrix such that $[\mathbf{K}] = K(x_i, x_j)$. The sequence $(\gamma_j, u_j)_{j \geq 1}$ is the eigenpair of $\mathbf{K}$. Then, the estimator for eigenfunctions $\varphi_j$'s and eigenvalues $\lambda_j$'s are

$$\varphi_j(x_i) \sim \sqrt{N} \, [u_j]_i \qquad \lambda_j \sim \frac{\gamma_j}{N}.$$

The $\sqrt{N}$ normalization for eigenfunction is to ensure that the squared integral of $\varphi_j$ on the observed samples is 1. That is,

$$\int_{\mathcal{X}} \varphi_j(x)\,\varphi_j(x)\,d\nu(x) \approx \frac{1}{N}\sum_{i=1}^{N}\varphi_j(x_i)\,\varphi_j(x_i) = \frac{1}{N}\sum_{i=1}^{N}\sqrt{N}[u_j]_i \cdot \sqrt{N}\,[u_j]_i = u_j^\mathsf{T} u_j = 1.$$

As for the eigenvalues, the proposed estimator is consistent. That is, $\gamma_j/N \to \lambda_j$ when $N \to \infty$ (Baker & Taylor, 1979, Theorem 3.4).

The estimator for eigenfunction only allows evaluation on points $x_1, \ldots, x_N$ used to solve the matrix eigenvalue equation. To evaluate the eigenfunction on arbitrary input, one can use a generalized Nyström-type estimator, defined as

$$\varphi_j(y) \sim \frac{\sqrt{N}}{\gamma_j}\sum_{i=1}^{N}K(y, x_i)\,[u_j]_i.$$

## D. Proof of Lower Bound

*Proof.* Let $\{\varphi_j\}_{j\in\mathbb{N}}$ be the eigenfunctions of $K$. That is,

$$\int_{\mathcal{X}} K(y, x)\,\varphi_i(x)\,dx = \lambda_i\,\varphi_i(y) \quad \forall i \in \mathbb{N}.$$

We now construct a hard distribution for the learner. Fix some $p \in (0, 1)$ and let $\xi_1, \xi_2, \ldots$ denote the sequence of pairwise independent random variables such that

$$\xi_j = \begin{cases} -\sqrt{1/p} & \text{with probability } \frac{p}{2} \\ 0 & \text{with probability } 1 - p\ . \\ \sqrt{1/p} & \text{with probability } \frac{p}{2} \end{cases}$$

Given such sequence, define a function $v$ such that

$$v(\cdot) = \sum_{j=1}^{\infty}\sqrt{\lambda_j}\,\xi_j\,\varphi_j(\cdot).$$

Note that

$$\|v\|_{L^2} = \sum_{j=1}^{\infty}\lambda_j\xi_j^2 < \infty$$

as $\sup_{j\in\mathbb{N}}|\xi_j|^2 \leq 1/p$ and $\sum_{j=1}^{\infty}\lambda_j < \infty$. Thus, $v$ is a random element in $L^2(\mathcal{X})$. Let $\mu$ denote the probability measure over $L^2(\mathcal{X})$ induced by the random sequence $\{\xi_j\}_{j\in\mathbb{N}}$. It is easy to see that $\mathbb{E}[v(x)] = 0$ for each $x \in \mathcal{X}$. Moreover, for every $x, y \in \mathcal{X}$, we have

$$\begin{aligned}
\mathbb{E}[v(x)\,v(y)] &= \mathbb{E}\left[\left(\sum_{j=1}^{\infty}\sqrt{\lambda_j}\,\xi_j\,\varphi_j(x)\right)\left(\sum_{j=1}^{\infty}\sqrt{\lambda_j}\,\xi_j\,\varphi_j(y)\right)\right] \\
&= \mathbb{E}\left[\sum_{j=1}^{\infty}\lambda_j\,\xi_j^2\varphi_j(x)\,\varphi_j(y) + 2\sum_{i<j}\sqrt{\lambda_i\lambda_j}\,\xi_i\xi_j\,\varphi_i(x)\,\varphi_j(y)\right] \\
&= \sum_{j=1}^{\infty}\lambda_j\,\mathbb{E}[\xi_j^2]\,\varphi_j(x)\,\varphi_j(y) \\
&= \sum_{j=1}^{\infty}\lambda_j\varphi_j(x)\,\varphi_j(y) \\
&= K(y, x),
\end{aligned}$$

where the final equality holds due to Mercer's theorem and the convergence is uniform over $x, y \in \mathcal{X}$. Therefore, we have shown that $\mu \in \mathcal{P}(K)$. Let $\sigma := \{\sigma_j\}_{j \geq 1}$ be a sequence of iid random variables such that $\sigma_j \sim \text{Uniform}(\{-1, 1\})$. Fix $c > 0$ and for each such $\sigma \in \{-1, 1\}^{\mathbb{N}}$, define

$$\mathcal{F}_\sigma := c \sum_{j=1}^{\infty} \sigma_j \, \varphi_j \otimes \varphi_j.$$

For each $m \in \mathbb{N}$, we will show that

$$\mathbb{E}_\sigma \left[ \mathbb{E}_{v_{1:n} \sim \mu^n} \left[ \mathbb{E}_{v \sim \mu} \left[ \left\| \widehat{\mathcal{F}}_n(v) - \mathcal{F}_\sigma(v) \right\|_{L^2}^2 \right] \right] \right] \geq \frac{c^2}{2} \sum_{j=1}^{m} \lambda_j.$$

Since this holds in expectation, using the probabilistic method, there must be a $\sigma^\star$ such that

$$\mathbb{E}_{v_{1:n} \sim \mu^n} \left[ \mathbb{E}_{v \sim \mu} \left[ \left\| \widehat{\mathcal{F}}_n(v) - \mathcal{F}_{\sigma^\star}(v) \right\|_{L^2}^2 \right] \right] \geq \frac{c^2}{2} \sum_{j=1}^{m} \lambda_j.$$

Noting that $\|\mathcal{F}_{\sigma^\star}\|_{\text{op}} = c$ completes our proof. The rest of the proof will establish this inequality.

Since $\{\varphi_j\}_{j \in \mathbb{N}}$ is the orthonormal bases of $L^2(\mathcal{X})$, Parseval's identity implies that

$$\left\| \widehat{\mathcal{F}}_n(v) - \mathcal{F}_\sigma(v) \right\|_{L^2}^2 = \sum_{j=1}^{\infty} \left| \left\langle \widehat{\mathcal{F}}_n(v) - \mathcal{F}_\sigma(v), \varphi_j \right\rangle \right|^2 = \sum_{j=1}^{\infty} \left| \left\langle \widehat{\mathcal{F}}_n(v), \varphi_j \right\rangle - \langle \mathcal{F}_\sigma(v), \varphi_j \rangle \right|^2.$$

Recall that $\mathcal{F}^\star(\varphi_j) = c \sigma_j \varphi_j$, where $\mathcal{F}^\star$ is the adjoint of $\mathcal{F}$. Thus, for any $v \sim \mu$, we have

$$\langle \mathcal{F}(v), \varphi_j \rangle = \langle v, \mathcal{F}^\star(\varphi_j) \rangle = \langle v, c \sigma_j \varphi_j \rangle = c \sigma_j \sqrt{\lambda_j} \, \xi_j,$$

which subsequently implies

$$\left\| \widehat{\mathcal{F}}_n(v) - \mathcal{F}_\sigma(v) \right\|_{L^2}^2 = \sum_{j=1}^{\infty} \left| \left\langle \widehat{\mathcal{F}}_n(v), \varphi_j \right\rangle - c \sigma_j \sqrt{\lambda_j} \, \xi_j \right|^2.$$

Using this fact, we can write

$$\mathbb{E}_\sigma \left[ \mathbb{E}_{v_{1:n} \sim \mu^n} \left[ \mathbb{E}_{v \sim \mu} \left[ \left\| \widehat{\mathcal{F}}_n(v) - \mathcal{F}_\sigma(v) \right\|_{L^2}^2 \right] \right] \right]$$

$$= \mathbb{E}_\sigma \left[ \mathbb{E}_{v_{1:n} \sim \mu^n} \left[ \mathbb{E}_{v \sim \mu} \left[ \sum_{j=1}^{\infty} \left| \left\langle \widehat{\mathcal{F}}_n(v), \varphi_j \right\rangle - c \sigma_j \sqrt{\lambda_j} \, \xi_j \right|^2 \right] \right] \right]$$

$$= \mathbb{E}_{v_{1:n} \sim \mu^n} \left[ \mathbb{E}_{v \sim \mu} \left[ \mathbb{E}_\sigma \left[ \sum_{j=1}^{\infty} \left| \left\langle \widehat{\mathcal{F}}_n(v), \varphi_j \right\rangle - c \sigma_j \sqrt{\lambda_j} \, \xi_j \right|^2 \right] \right] \right].$$

In the final step, we changed the order of integration. Note that drawing $n$ samples of $v_1, \ldots, v_n$ and drawing $\sigma$ can be done in any order, as they are interchangeable. Finally, the draw of $v \sim \mu$ occurs during the test phase, independent of the previously drawn samples $v_{1:n}$ and $\sigma$.

Next, let $E_{n,m}$ denote the event such that

$$\langle v_i, \varphi_j \rangle = 0 \qquad \forall 1 \leq i \leq n \text{ and } 1 \leq j \leq m.$$

Then, we will lowerbound

$$\mathbb{E}_{v \sim \mu} \left[ \mathbb{E}_\sigma \left[ \sum_{j=1}^{\infty} \left| \left\langle \widehat{\mathcal{F}}_n(v), \varphi_j \right\rangle - c \sigma_j \sqrt{\lambda_j} \, \xi_j \right|^2 \right] \right]$$

conditioned on the event $E_{n,m}$. First, note that

$$\underset{v \sim \mu}{\mathbb{E}} \left[ \underset{\sigma}{\mathbb{E}} \left[ \sum_{j=1}^{\infty} \left| \left\langle \widehat{\mathcal{F}}_n(v), \varphi_j \right\rangle - c\, \sigma_j \sqrt{\lambda_j}\, \xi_j \right|^2 \right] \right] \geq \underset{v \sim \mu}{\mathbb{E}} \left[ \sum_{j=1}^{m} \underset{\sigma}{\mathbb{E}} \left[ \left| \left\langle \widehat{\mathcal{F}}_n(v), \varphi_j \right\rangle - c\, \sigma_j \sqrt{\lambda_j}\, \xi_j \right|^2 \right] \right]$$

$$\geq \underset{v \sim \mu}{\mathbb{E}} \left[ \sum_{j=1}^{m} \left( \underset{\sigma}{\mathbb{E}} \left| \left\langle \widehat{\mathcal{F}}_n(v), \varphi_j \right\rangle - c\, \sigma_j \sqrt{\lambda_j}\, \xi_j \right| \right)^2 \right],$$

where the final step uses Jensen's inequality. Next, we use the fact that when the event $E_{n,m}$ occurs, the learner has no information about $\sigma_1, \ldots, \sigma_m$. This is because the input data shows no variation along the directions spanned by $\varphi_1, \ldots, \varphi_m$. Given that $\mathcal{O}$ is the perfect oracle for $\mathcal{F}_\sigma$, any information provided by the oracle $\mathcal{O}$ must be independent of how $\mathcal{F}_\sigma$ operates on the subspace spanned by $\varphi_1, \ldots, \varphi_m$. Specifically, for every $1 \leq i \leq n$ and $1 \leq j \leq m$, the output of the oracle $\mathcal{O}(v_i)$ must be independent of $\sigma_j$. If this condition holds, then the estimator $\widehat{\mathcal{F}}_n$ must also be independent of $\sigma_1, \ldots, \sigma_m$. Thus, conditioned on the event $E_{n,m}$, for any $1 \leq j \leq m$, we have

$$\underset{\sigma}{\mathbb{E}} \left| \left\langle \widehat{\mathcal{F}}_n(v), \varphi_j \right\rangle - c\sigma_j \sqrt{\lambda_j}\, \xi_j \right| = \mathbb{E} \left[ \underset{\sigma_j}{\mathbb{E}} \left[ \left| \left\langle \widehat{\mathcal{F}}_n(v), \varphi_j \right\rangle - c\sigma_j \sqrt{\lambda_j}\, \xi_j \right| \; \middle| \; \sigma \backslash \{\sigma_j\} \right] \right]$$

$$= \mathbb{E} \left[ \frac{1}{2} \left| \left\langle \widehat{\mathcal{F}}_n(v), \varphi_j \right\rangle - c \sqrt{\lambda_j}\, \xi_j \right| + \left| \left\langle \widehat{\mathcal{F}}_n(v), \varphi_j \right\rangle + c \sqrt{\lambda_j}\, \xi_j \right| \right]$$

$$\geq \frac{1}{2} \left| 2\, c\, \sqrt{\lambda_j} \xi_j \right|$$

$$= \left| c \sqrt{\lambda_j}\, \xi_j \right|.$$

The first equality uses the fact that conditioned on $\sigma \backslash \{\sigma_j\}$, the function $\widehat{\mathcal{F}}_n(v)$ is independent of $\sigma_j$. Thus, conditioned on the event $E_{n,m}$, we have shown that

$$\underset{v \sim \mu}{\mathbb{E}} \left[ \underset{\sigma}{\mathbb{E}} \left[ \sum_{j=1}^{\infty} \left| \left\langle \widehat{\mathcal{F}}(v), \varphi_j \right\rangle - c\, \sigma_j \sqrt{\lambda_j}\, \xi_j \right|^2 \right] \right] \geq \underset{v \sim \mu}{\mathbb{E}} \left[ \sum_{j=1}^{m} c^2 \lambda_j \xi_j^2 \right] = c^2 \sum_{j=1}^{m} \lambda_j \mathbb{E}[\xi_j^2] = c^2 \sum_{j=1}^{m} \lambda_j.$$

Therefore, our overall lowerbound is

$$\underset{v_{1:n} \sim \mu^n}{\mathbb{E}} \left[ \underset{v \sim \mu}{\mathbb{E}} \left[ \underset{\sigma}{\mathbb{E}} \left[ \sum_{j=1}^{\infty} \left| \left\langle \widehat{\mathcal{F}}(v), \varphi_j \right\rangle - c\, \sigma_j \sqrt{\lambda_j}\, \xi_j \right|^2 \right] \right] \right] \geq c^2 \left( \sum_{j=1}^{m} \lambda_j \right) \mathbb{P}\left[ E_{n,m} \right] = c^2 (1-p)^{n \cdot m} \sum_{j=1}^{m} \lambda_j.$$

The final step uses the fact that $\mathbb{P}[E_{n,m}] = (1-p)^{n \cdot m}$. It now remains to pick $p$ to obtain the claimed lowerbound. Let us pick $p = \frac{1}{2mn}$. Then, we have $(1-p)^{mn} \geq 1/2$ as long as $n \geq 1$, yielding the lowerbound of

$$\frac{c^2}{2} \sum_{j=1}^{m} \lambda_j.$$

Since $m \in \mathbb{N}$ is arbitrary, our lowerbound holds for every fixed $m$. Noting that $\|\mathcal{F}_\sigma\|_{\mathrm{op}} = c$ for every $\sigma$ completes our proof. $\qquad \square$

## E. Experiments

This section presents additional experimental results using the same setup as described in Section 5. The results show that the Fourier Neural Operator (FNO) performs poorly with actively collected data. This is likely because the training data are not i.i.d. samples from the test distribution, requiring FNO to generalize out of distribution when trained on actively collected data.

### E.1. Poisson Equation

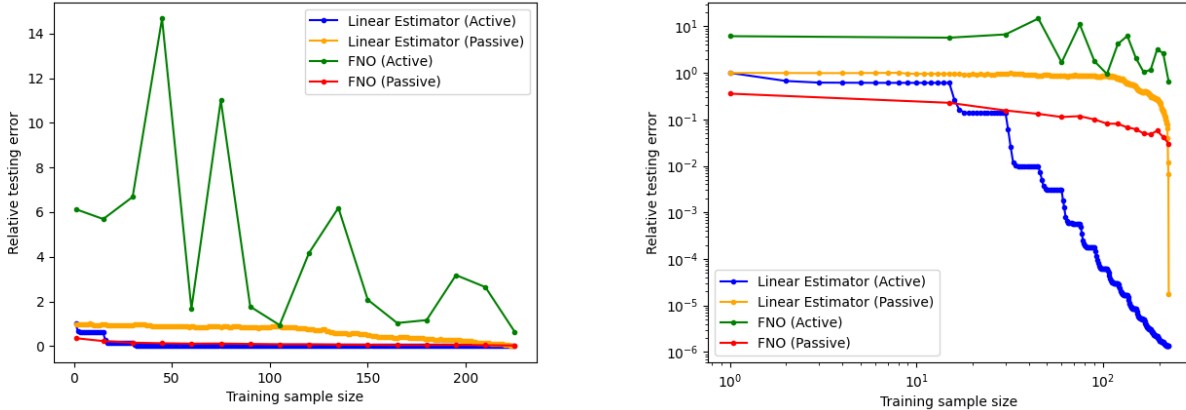

*Figure 5.* Error Plots for various estimators for Poisson Equation. The plot on the right shows the same plot in log scale.

### E.2. Heat Equation

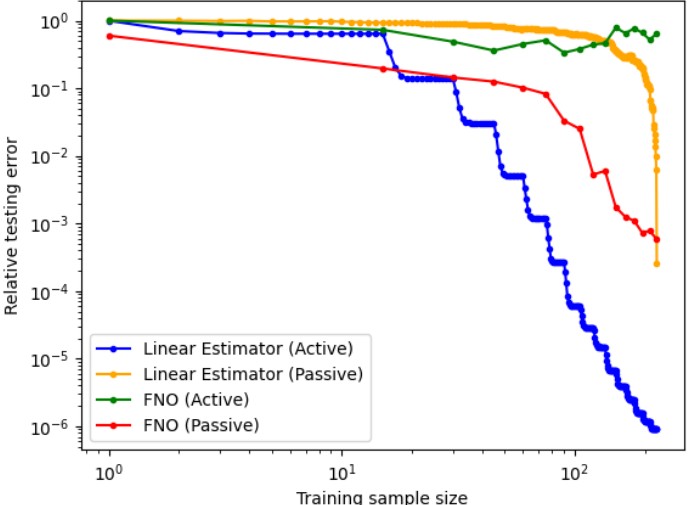

*Figure 6.* Error Plots for various estimators for Heat Equations in log-log scale.

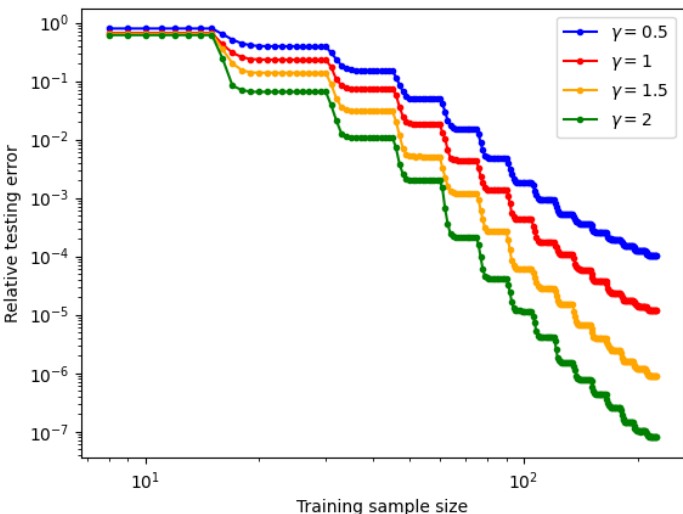

*Figure 7.* Convergence rate of the active linear estimator for Heat equation with actively collected data for different values of $\gamma$.

