# OpenReview forum: "On the Benefits of Active Data Collection in Operator Learning"
_ICML.cc/2025/Conference — ICML 2025 spotlightposter_

### Official Review · Reviewer_rbqN · 2025-03-02

**Overall Recommendation:** 4

**Summary:**

This paper studies the approximating solution operator of the PDE using active learning queries given the kernal. The paper includes an upperbound for an active learning setting that converges to an irreducible error with a larger number of queries. Further, the authors show a lower bound for passive learning. The paper also provides some numerical experiments using their estimators to learn Poisson and Heat Equations.

**Claims And Evidence:**

The submission provides both rigorous proof and numerical experiments.

**Essential References Not Discussed:**

N/A

**Experimental Designs Or Analyses:**

N/A

**Methods And Evaluation Criteria:**

Both the problem setting and the strategy of solving Feldhome integral for picking queries make sense.

**Other Comments Or Suggestions:**

N/A

**Other Strengths And Weaknesses:**

N/A

**Questions For Authors:**

The authors used active learning with membership queries for the setting, which seems most natural. However, I wonder if the authors can discuss what other types of queries they think might be interesting for PDE approximation if they have any in mind.

**Relation To Broader Scientific Literature:**

This paper proposed a new setting for operative learning and provided an algorithm with theoretical guarantees.

**Theoretical Claims:**

I went through the proof for Theorem 3.1 in Appendix A. I did not find any issues.

---

> ### Author Rebuttal · Authors · 2025-03-30
>
> We thank the reviewer for noting that our work provided rigorous proofs and numerical experiments.
>
> * **“I wonder if the authors can discuss what other types of queries they think might be interesting for PDE approximation if they have any in mind.”**
> One natural direction would be to explore a pool-based active learning setting, where the learner selects inputs from a large (possibly infinite) unlabeled pool drawn from a fixed distribution $ \mu $, rather than querying arbitrary inputs. This may be more appropriate in scenarios where labeling requires running physical experiments, making arbitrary queries impractical. However, in the context of PDE surrogate modeling, where data is generated synthetically, we believe the membership-type query model we adopt that allows arbitrary input queries is reasonable.

---

### Official Review · Reviewer_1DAM · 2025-03-11

**Overall Recommendation:** 4

**Summary:**

The authors study active learning of linear operator motivated by the context of approximating the solution operator of linear PDEs, a setting where the user may "manufacture" a small number of queries to give to a PDE oracle in order to approximate the underlying linear solution operator.

In slightly more detail, the authors consider the problem of learning an infinite dimensional linear operator T over the set of centered, square-integrable stochastic processes with known covariance kernel K arising as the solution function of some PDE $Lu=f$ over domain $X \subset \mathbb{R}^d$. The authors work in the (noisy) membership query model $\mathcal{O}$, meaning they are given access to an oracle $\mathcal{O}$ which on query a function $u$ over the domain, outputs a function $\mathcal{O}(u)$ which is within $\varepsilon$ in L_2 of the true solution $T(u)$. This is meant to model the output of a query made to a numerical PDE solver which may have some small error $\varepsilon$ based on the discretization level. The learner's goal is to output an operator $\hat{T}$ minimizing the $L_2$ error from the true operator $T$.


The authors propose a query algorithm for this problem which simply queries the top n eigenvectors $\{\phi_i\}$ of K and outputs the estimator $\hat{T} = \sum\limits_{i=1}^n \mathcal{O}(\phi_i) \otimes \phi_i$. They prove this estimator has error at most:

$\varepsilon^2\sum_{i=1}^n \lambda_i + ||T||\sum_{i=n+1}^\infty \lambda_i$

which is vanishing as the number queries $n \to \infty$ and the measurement error $\varepsilon \to 0$.

The authors then explicitly calculate the error rate of their algorithm for several classical settings (fractional inverse of shifted Laplacian, RBF Kernel, Brownian motion...), and show that for many natural parameter settings their active algorithm has significantly improved error rate over passive methods which draw samples from the underlying process rather than actively querying specially constructed functions and typically have at best inverse linear error decay. For instance, for RBF kernels the authors show *exponential* decay of error in 1-D, and better than polynomial decay for any fixed dimension $d$.

The authors also show that if one only restricts the unknown stochastic process to be generated by some fixed K, there is always an underlying distribution that has *non-vanishing* rate for passive learning, while their active strategy always has rate tending to 0.

Finally, the authors perform a number of experiments showing the advantage of their active procedure over its classical passive counterpart, showing empirical success even in parameter settings where their formal guarantees are not known to hold.

**Claims And Evidence:**

Yes.

**Essential References Not Discussed:**

Not to my knowledge.

**Experimental Designs Or Analyses:**

No.

**Methods And Evaluation Criteria:**

Yes.

**Other Comments Or Suggestions:**

GP denotes Gaussian Process is stated well after the first time GP is used.

**Other Strengths And Weaknesses:**

The membership query model is very infrequently studied in active learning due to generally being considered unrealistic (e.g. one cannot really give a synthesized data point to a human being to label and expect a coherent response). One additional strength of this work is identifying (linear) PDE solving as a potential application where the membership query model really is plausible, and provides substantially improved error rates over its passive counterpart (or in fact even over the standard active pool-based model, which I believe the authors lower bound also holds for).

From a learning standpoint, it is a bit disappointing that the proposed algorithm relies so strongly on knowing the covariance kernel K, and I am not convinced the assumption is all that reasonable (though the authors make the case it is a standard assumption in prior work on operator learning). Knowing K allows one to work directly with its (known) eigenfunctions, sort of immediately unlocking "PCA" style techniques with no need for estimation. Often in statistical learning one would expect to have to learn some of the underlying geometry to do something like this which is avoided here.

EDIT: I am largely happy with the authors' response regarding known vs unknown K, and suggest they include a formalization of the discussion in the next version.

**Questions For Authors:**

Can one show knowledge of K is necessary to achieve fast rates, at least in some settings?

**Relation To Broader Scientific Literature:**

The key contribution of this paper is to establish *active learning* rates for linear operator estimation. Prior work focused only on the passive setting, but as the authors reasonably argue, in many cases we have access to an approximate PDE solver which we can feed queries of our own design. They give a very simple query algorithm achieving substantially better error rates than can be achieved passively so long as one can compute the eigenfunctions of the known covariance kernel.

**Theoretical Claims:**

The proof sketch provided in the main body is convincing of their main theorem.

---

> ### Author Rebuttal · Authors · 2025-03-30
>
> We thank the reviewer for their thoughtful feedback. We address their concerns below.
>
> *   **``On the assumption of a known covariance kernel $K$:"** We agree that assuming knowledge of the kernel $K$ may seem unnatural from the perspective of classical statistical learning. However, in operator learning, this assumption is often more reasonable, as data is typically generated synthetically via simulations, where the input distribution is controlled by the user. For example, common covariance operators like $\alpha(-\nabla^2 + \beta I)^{-\gamma}$ on a periodic domain have known eigenfunctions—specifically, complex exponentials $e^{2\pi i m \cdot x}$, with the parameters $(\alpha, \beta, \gamma)$ only affecting the eigenvalues. Since our estimator relies only on the eigenfunctions, it effectively assumes that inputs are represented in a known basis determined by the domain geometry (e.g., Fourier basis on the torus, spherical harmonics on the sphere), not on the detailed spectral properties of the kernel. That said, this separation breaks down for kernels like the RBF (squared exponential), where the scale influences both eigenvalues and eigenfunctions. In such cases, assuming knowledge of the eigenfunctions is indeed more restrictive.
>
>
> * In addition, from a learning-theoretic perspective, assuming knowledge of the kernel $K$ is arguably without loss of generality. In active learning, it's common to assume access to an unlimited pool of unlabeled samples $v_1, \ldots, v_m \sim_{\text{iid}} \mu$, where $\mu \in \mathcal{P}(K)$, and focus on minimizing label complexity—the number of labeled samples requested (see Hanneke, 2014). This aligns with our setting, where labeling (e.g., solving a PDE) is the primary cost. Given such unlabeled samples, one can estimate the covariance operator as:
> $$ \Sigma\_m = \frac{1}{m-1} \sum_{i=1}^m (v_i - \overline{v}\_m) \otimes (v_i - \overline{v}\_m),  $$
> where $\overline{v}\_m = \frac{1}{m} \sum_{i=1}^m v_i.$
> Since $\mathbb{E}[||v_i||^2] < \infty$, Theorem 8.1.2 of Hsing and Eubank (2015) guarantees that $\Sigma_m \to \Sigma$ almost surely in Hilbert-Schmidt norm, where $\Sigma$ is the integral operator associated with $K$. While our work assumes $\Sigma$ has a finite trace norm, this is not required to recover its eigenfunctions: convergence in Hilbert-Schmidt norm suffices for accurate top $n$ eigenfunctions approximation. Thus, assuming access to the eigenfunctions of $K$ is reasonable in theory, even if it may be computationally demanding in practice. We also note that this does not contradict known lower bounds in pool-based active learning, since our approach queries the oracle on estimated eigenfunctions of $\Sigma_m$, which are not i.i.d. samples from $\mu$.
>
>
> * **``Can one show knowledge of $K$ is necessary to achieve fast rates, at least in some settings?"**
> Yes, some knowledge of the kernel $K$ is necessary to achieve fast rates for certain estimators. Earlier, we showed that if the learner has access to unlabeled samples from a distribution $\mu \in \mathcal{P}(K)$, then $K$ can be estimated from data, making the assumption of known $K$ natural. Here, we consider the more adversarial setting where the learner has no access to such samples. Suppose the learner selects $n$ inputs $v_1, \ldots, v_n$ using any active strategy and receives exact labels $w_i = \mathcal{F}(v_i)$. Let $\widehat{\mathcal{F}}\_n$ be the resulting estimator. As is common with many linear estimators, we assume $\widehat{\mathcal{F}}\_n(v) = 0$ for any $v$ outside the span of $v_1, \ldots, v_n$. This models abstention outside the observed subspace. Since learner had no knowledge of $K$, we can construct a kernel
> $$
> K(x, y) = \sum_{j=1}^M \lambda_j \varphi_j(x) \varphi_j(y),
> $$
> where the orthonormal functions $\varphi_j$ are chosen to be orthogonal to the span of the learner's queries. This is always possible in infinite-dimensional $L^2(\mathcal{X})$. We then define $\mu \in \mathcal{P}(K)$ as the law of a Gaussian process with kernel $K$.
> Since the entire support of $\mu$ lies outside the learner’s observed subspace, the estimator cannot generalize. In particular,
> $$
> \mathbb{E}[||\widehat{\mathcal{F}}\_n(v) - \mathcal{F}(v)||^2] \geq \sum_{j=1}^M \lambda_j ||\mathcal{F}(\varphi_j)||^2.
> $$
> If $\mathcal{F}$ is not finite-rank, we can always find some $\varphi_\ell$ with $||\mathcal{F}(\varphi_\ell)|||^2 \geq c > 0$. By setting $\lambda_\ell = 1$ and choosing the rest of the $\lambda_j$ to satisfy a trace constraint, the lower bound remains constant, independent of $n$. In short, if the learner lacks both knowledge of $K$ and access to samples from $\mu \in \mathcal{P}(K)$, there exist kernels that force all probability mass outside the learner’s span, leading to a non-vanishing error regardless of how data is collected.
>
>
> [1] Hanneke, Steve.``Theory of active learning." Foundations and Trends in Machine Learning 7.2-3 (2014).

---

### Official Review · Reviewer_E2pE · 2025-03-12

**Overall Recommendation:** 4

**Summary:**

The authors study the problem of learning a bounded, linear operator through active learning with the assumption that the input functions are drawn from a mean 0 distribution with a known continuous covariance kernel, $K$.

Their main contribution is a deterministic strategy, which involves solving the Fredholm integral equation to obtain eigenfunctions which are  selected as input functions to query the oracle (eg. a PDE solver). They show polynomial (for fractional inverse), exponential (RBF), 1/n (Brownian motion) convergence rates for some common kernels.

They motivate their result further with a lower bound of $\|\mathcal{F}\|_{op}^2 \sum_{j=1}^m \lambda_j/2$, and show the effectiveness of their method with numerical experiments on the Poisson and Heat Equations by comparing with passive strategies using a linear estimator and Fourier Neural Operators.

**Claims And Evidence:**

This paper is mathematically rigorous, providing guarantees for all their results, including theorem 3.1, 4.2 and also showing the measurability conditions to ensure a meaningful definition of $\mathcal P(\mathcal X)$, a probability distribution over $L^2(\mathcal X)$ induced by the stochastic process with kernel $K$.

**Essential References Not Discussed:**

I don't think there are.

**Experimental Designs Or Analyses:**

Their experiment design sounds well motivated.

**Methods And Evaluation Criteria:**

Their evaluation framework, of considering error convergence rate in terms of eigen value decay of $K$ makes sense, and experimentally, both the problems they test on Poisson Equation and Heat Equation are relevant applications.

**Other Comments Or Suggestions:**

none.

**Other Strengths And Weaknesses:**

Strengths:
* They provide strong theoretical guarantees, support their method with a lower bound on passive learning, and provide empirical evidence.

Weaknesses:
* While this is a theoretical study, it would be an interesting next step to see their method on a broader range of PDEs.
* It would be helpful to include an empirical comparison with other active learning methods from the literature, if any exist.
* It would also be good for the reader to see some intuition for the lower bound construction in the main paper instead of the appendix.

**Questions For Authors:**

Your method selects the top eigenfunctions of $K$, and ultimately you leverage the spectral decay of $K$ to achieve fast convergence rates. However it requires exactly solving the Fredholmer integral or numerically approximating it. Have you considered a randomized sampling method, where input functions could be sampled with probabilities inversely proportional to the Christoffel function? Maybe it could be computationally cheaper?

**Relation To Broader Scientific Literature:**

Their method is in a similar setting as Kovachki et al., 2023, but sample efficient. Their setting is for bounded linear operators, while Lipschitz operators, ones with known SVD has been studied by y de Hoop et al. (2023), Subedi & Tewari (2024), Liu et al. (2024). Their work is a novel active learning method in this specific setting for a known covariance continuous kernel, which goes beyond passive learning. The sample complexity for commonly used kernels used in modeling is also a meaningful advancement.

**Theoretical Claims:**

I did not check the proofs in detail.

---

> ### Author Rebuttal · Authors · 2025-03-30
>
> We thank the reviewer for their thoughtful comment and noting that our work provides strong theoretical guarantees with empirical evidence. We address reviewer's concern below.
>
>
> *   **``While this is a theoretical study, it would be an interesting next step to see their method on a broader range of PDEs."**
>  We agree with the reviewer that generating our approach to handle potentially non-linear solution operators is an important future direction. We view our work as an important first step that lays a theoretical foundation for future works on active data collection in operator learning.
>
> *   **``It would be helpful to include an empirical comparison with other active learning methods from the literature, if any exist."**
>  Currently, there is no widely accepted active learning baseline for operator learning. Unlike the passive setting, where empirical risk minimization on i.i.d. samples is standard, active learning strategies are typically problem-specific and do not scale well to the infinite-dimensional setting.  For example, uncertainty sampling and Bayesian algorithms will both require careful extensions to infinite dimensions.
>
> * **"It would also be good for the reader to see some intuition for the lower bound construction in the main paper instead of the appendix."**
>  We will include a short proof sketch of our lower bound construction highlighting key ideas in the main text of the final version of the paper.
>
> * **``Have you considered a randomized sampling method, where input functions could be sampled with probabilities inversely proportional to the Christoffel function? Maybe it could be computationally cheaper?"**  We thank the reviewer for this thoughtful suggestion. Indeed, computing the top eigenfunctions of the kernel requires solving (or approximating) the Fredholm integral equation, which can be computationally intensive for domains with complex geometry. We had not considered Christoffel function-based randomized sampling, but it seems like a promising approach especially when $d$ is large. We appreciate this idea and will keep it in mind for our ongoing and future works.

---

### Official Review · Reviewer_fV81 · 2025-03-16

**Overall Recommendation:** 3

**Summary:**

The authors consider active data collection in operator learning with distributions, induced by a stochastic process, over function spaces. They obtained an upper bound for active data collection by spectral techniques and a lower bound for passive data collection, which shows the benefit of the active approach. Concrete examples of various stochastic processes are presented, and experiments are conducted to support the theoretical findings.

**Claims And Evidence:**

The claims are supported by rigorous roofs and experiments.

**Essential References Not Discussed:**

No.

**Experimental Designs Or Analyses:**

Yes.

**Methods And Evaluation Criteria:**

Yes.

**Other Comments Or Suggestions:**

Given the numerous probability measures discussed in the paper, it is better to specify which probability measure is associated with the L^2 norm to ensure mathematical clarity.

**Other Strengths And Weaknesses:**

The topic and the problem setting in the paper are indeed intriguing. However, the theoretical results presented seem quite limited in their mathematical contribution.

**Questions For Authors:**

1. Provide specific examples in operator learning applications which meet the setting in 2.3.
2. Explain why "we typically have \epsilon ~ N^-s" above Section 3: if we consider \epsilon as an approximation error, the convergence rate here doesn't suffer from the curse of dimensionality. More important, without assumptions on function spaces (just L^2 space), a polynomial rate cannot be achieved in operator learning.

**Relation To Broader Scientific Literature:**

The prior studies focus on both approximation and sampling complexity of operator learning. The active data collection proposed in the paper is to investigate the data sampling and improve statistical efficiency in operator learning. Since operators are defined on more complex topological spaces than Euclidean spaces, the convergence rate is much slower compared to their Euclidean counterparts. The active data collection method propose an interesting idea to improve it.

**Theoretical Claims:**

Yes. Theorem 3.1.

---

> ### Author Rebuttal · Authors · 2025-03-30
>
> We thank the reviewer for their thoughtful comments and for noting that the topic and the problem setting in the paper are intriguing. Below we respond to the main concerns and questions.
>
> *  **On the $L^2$ norm and associated probability measure:**
> We thank the reviewer for pointing this out. As discussed in Section 2.1, the $L^2$ norm is taken with respect to the base measure $\nu$. This is usually Lebesgue but can be more general such as Gaussian weighting. We will make this more explicit in the revision to ensure clarity.
>
> * **"Q1: Provide examples where the assumptions in Section 2.3 apply."**  We believe this is a standard framework commonly adopted in operator learning works [1,2]. While Assumption 2.1 is not always explicitly stated, it is typically implicit in the empirical evaluations of these methods. A more detailed discussion of such kernels can be found in [3]. By making this assumption explicit and combining it with active data collection, we are able to derive stronger guarantees—specifically, uniform bound over all distributions in the family $\mathcal{P}(K)$. In contrast, existing theoretical results usually provide guarantees only for the specific distribution used to generate the training data.
>
> * **Q2: Explain why ``we typically have $\epsilon \sim N^{-s}$" above Section 3: if we consider $\epsilon$ as an approximation error, the convergence rate here doesn't suffer from the curse of dimensionality. More important, without assumptions on function spaces (just $L^2$ space), a polynomial rate cannot be achieved in operator learning.**
> The reviewer is correct that achieving polynomial approximation rates is generally non-trivial in operator learning. However, we would like to clarify that $ \varepsilon $ in our setting does not denote the approximation error of the operator class. Rather, it corresponds to the error in training data due to the error of the PDE solver (i.e., the oracle $ \mathcal{O} $) used to generate training data.
> That said, if the reviewer is referring to the error from approximating functions in $ L^2 $ using a truncated basis (e.g., Fourier series), we agree that the commonly stated rate $ N^{-s} $ can be somewhat misleading. For functions defined on a $ d $-dimensional domain, this rate holds only when all Fourier modes $ k \in \mathbb{Z}^d $ satisfying $ |k|_{\infty} \leq N $ are included. If instead only the first $ N $ modes (in total number) are used, the convergence rate typically becomes $ N^{-s/d} $ for functions with $ s $-degree smoothness. This rate suffers from a curse of dimensionality as the reviewer notes. In our setting, this smoothness assumption is justified: the PDE solver is only applied to the eigenfunctions of the kernel $ K $, which are typically smooth when $ K $ is sufficiently regular. We appreciate the reviewer’s sharp observation and will update the discussion in the final version to clarify this point.
>
>
> [1] Li, Zongyi, et al. ``Fourier Neural Operator for Parametric Partial Differential Equations." International Conference on Learning Representations. 2021.
>
> [2] Kovachki, Nikola, et al. "Neural operator: Learning maps between function spaces with applications to pdes." Journal of Machine Learning Research 24.89 (2023): 1-97.
>
> [3] Boullé, Nicolas, and Alex Townsend. "A mathematical guide to operator learning." arXiv preprint arXiv:2312.14688 (2023).

---

### Official Review · Reviewer_UK7y · 2025-03-16

**Overall Recommendation:** 4

**Summary:**

This paper proposes an active learning method to learn bounded linear operators from data. This method selects input functions based on the eigenfunctions of the covariance kernel, leading to faster convergence rates. The paper establishes minimax optimal error bounds, showing that active learning can outperform passive learning, especially when the kernel's eigenvalues decay rapidly. Numerical experiments on PDEs like Poisson and Heat equations support the theoretical findings.

**Claims And Evidence:**

Yes, the paper supports its claims with thorough theoretical analysis and well-designed experiments. It provides detailed proof for the convergence rates of both passive and active learning strategies, clearly demonstrating the advantages of active learning under certain conditions. Additionally, the experimental results on PDE benchmarks align with the theoretical findings, reinforcing the validity of the claims. Overall, the evidence presented is both convincing and comprehensive.

**Essential References Not Discussed:**

No.

**Experimental Designs Or Analyses:**

Yes, the paper employs both the proposed linear operator estimator and FNO (Fourier Neural Operator) with passive learning as baselines. The experimental design is sound, as it systematically compares these approaches to demonstrate the benefits of active learning.

Additional Questions to the Authors:

Q1: Could FNO benefit from using the actively selected training data chosen by the proposed linear estimator?
Q2: The number of training samples in the experiments appears to be chosen empirically. Can they be potentially guided by Theorem 3.1 or any theoretical criteria?
Q3:  The paper focuses on comparing convergence rates rather than absolute final accuracy. Given that FNO has greater expressive capacity, is it possible that with a large enough dataset, FNO might eventually match or outperform the linear estimator?

**Methods And Evaluation Criteria:**

Yes.

**Other Comments Or Suggestions:**

None.

**Other Strengths And Weaknesses:**

Strengths:
The paper is well-written and offers a clear, rigorous theoretical framework comparing passive and active learning in operator learning. It provides significant insights by establishing minimax optimal error bounds and highlighting when active learning is beneficial. The use of eigenfunctions in the data selection strategy is elegant and well-motivated.

Weaknesses:
While theoretically strong, the experimental section is limited to relatively simple PDE cases, and it’s unclear how well the approach scales to more complex, real-world scenarios or nonlinear operators. Some practical implications remain unexplored.

**Questions For Authors:**

See Experimental Designs Or Analyses.

**Relation To Broader Scientific Literature:**

This paper contributes to the literature on operator learning by providing sharp theoretical comparisons between passive and active learning strategies. It builds upon prior works using Gaussian processes and linear operators, extending them with new minimax optimal error bounds.

**Theoretical Claims:**

I have carefully reviewed the theoretical claims presented up to Section 3.2, including the setup, assumptions, upper bound and the linear estimator, and they appear sound and reasonable to me. However, I find it challenging to fully verify the correctness of the more technical proofs and theoretical results presented in the later sections, particularly those involving detailed operator norm bounds and minimax lower bounds, as they require more advanced mathematical rigor and deeper familiarity with the functional analysis tools used.

---

> ### Author Rebuttal · Authors · 2025-03-30
>
> We thank the reviewer for their encouraging and positive assessment, and for recognizing that our work offers significant insights by establishing minimax-optimal error bounds and clarifying when active learning is beneficial. We address the reviewer’s questions below.
>
>
> *  **``Q1: Could FNO benefit from using the actively selected training data chosen by the proposed linear estimator?"**   Interestingly, FNO performs poorly when trained on actively chosen data for the linear estimator. We include the error curve of FNO trained on these active samples in Appendix E.
>
> *  **``Q2: The number of training samples in the experiments appears to be chosen empirically. Can they be potentially guided by Theorem 3.1 or any theoretical criteria?"** Yes, in principle, our bound can guide the choice of sample size. Suppose we want to select $ n $ such that the reducible error term
> $$  || \mathcal{F} ||\_{\mathrm{op}}  \sum_{j > n} \lambda_j $$
> is at most some small  $\delta > 0 $. Assume the eigenvalues decay polynomially, i.e.,  $ \lambda_j \lesssim j^{-p} $ for some $ p > 1 $. Note that $p>1$ is required for the kernel to be Mercer. Then we have
> $$
> || \mathcal{F} ||\_{\text{op}} \sum_{j > n} \lambda_j \lesssim || \mathcal{F}||\_{\text{op}} \sum_{j > n} j^{-p} \lesssim ||\mathcal{F}||\_{\text{op}} n^{-p+1} \leq \delta,
> $$
>  as long as
> $$
> n \gtrsim \left( \frac{||\mathcal{F}||\_{\text{op}}}{\delta} \right)^{\frac{1}{p-1}}.
> $$
> For example, when $ p = 2 $, we recover the sample complexity corresponding to the standard fast rates.
>
> *  **``Q3: The paper focuses on comparing convergence rates rather than absolute final accuracy. Given that FNO has greater expressive capacity, is it possible that with a large enough dataset, FNO might eventually match or outperform the linear estimator?"**  As discussed above, when the eigenvalues decay polynomially at the rate $ \lambda_j \lesssim j^{-p} $ for some $ p > 1 $, the error of our estimator scales as
> $$
> \lesssim \frac{||\mathcal{F} ||\_{\text{op}}}{n^{p-1}}.
> $$
> In contrast, with i.i.d. samples, the best convergence rate achievable by FNO is
> $$
> \lesssim \frac{\text{(some notion of complexity of the FNO model class)}}{n}.
> $$
> For FNO model to capture $\mathcal{F}$, the resulting notion of complexity of the model class is generally $\geq || \mathcal{F} ||\_{\text{op}}$. Thus, while both rates decay to zero as $ n \to \infty $, our estimator achieves a faster convergence rate when $ p > 1 $. Therefore, for any given sample size $n$, our estimator should always have a smaller error than FNO in such cases.
>
> *  **``Weaknesses: While theoretically strong, the experimental section is limited to relatively simple PDE cases, and it’s unclear how well the approach scales to more complex, real-world scenarios or nonlinear operators. Some practical implications remain unexplored."**  Since our theoretical guarantees apply only to linear PDEs, we focus on standard linear PDEs in our experiments. That said, we agree with the reviewer that extending active learning approaches to nonlinear operators is essential for addressing complex real-world scenarios. We view our work as an important first step that lays a theoretical foundation for future research in that direction.

---

### Official Review · Reviewer_XHLU · 2025-03-19

**Overall Recommendation:** 3

**Summary:**

This paper is in the general area of using AI for PDE. Its goal is to minimize the input-output pairs needed to train such an AI model. The paper proves a new bound on the sample complexity. The results show that the proposed method have arbitrarily fast error convergence rates with sufficiently rapid eigenvalue decay of the covariance kernels.

**Claims And Evidence:**

The claims are backed up with theoretical analysis and experiments.

**Essential References Not Discussed:**

N/A

**Experimental Designs Or Analyses:**

The evaluation results look reasonable.

**Methods And Evaluation Criteria:**

I do not have theoretical backgrounds to evaluate the theoretical aspects of the paper, so my comments will be focused on the experimental setup and results.

One issue is that the paper is that the functions they have studied are pretty simple and does not reflect the complexity in real science applications. The authors only evaluate two equations. I also don't know whether the problem size is enough. I would imagine for large problem size (e.g., complex functions), it would take much more samples intelligently in order to recover the function. I suggest the authors think more thoroughly about how to empirically evaluate the methods. I think the problem is interesting only at a large scale. If an algorithm can converge using 30 samples only, a human probably can manually write down the function directly. Does the grid size matter in the evaluation?

Another question is about the experiment setup, what are the set of functions the proposed method use to approximate the original function is not clear.

**Other Comments Or Suggestions:**

N/A

**Other Strengths And Weaknesses:**

N/A

**Questions For Authors:**

Do you have evaluation results for more complex functions?

Does the grid size matter in the evaluation?

Is there any baseline for comparison in the active learning field that can be used directly for operator learning?

**Relation To Broader Scientific Literature:**

I think the overall direction is very interesting and can have significant implications.

**Theoretical Claims:**

I didn't check the correctness of the theoretical proofs.

---

> ### Author Rebuttal · Authors · 2025-03-30
>
> We thank the reviewer for their valuable feedback and for recognizing that the overall direction of our work is interesting and potentially impactful. Below, we address the reviewer’s questions and concerns.
>
>
> * We agree with the reviewer that, in its current form, the scope of our work does not fully capture the complexity of real-world PDE problems. Regarding the comment on the small number of samples needed to recover the operator, we note that this efficiency is possibly due to the *linearity* of the underlying operator and the use of actively chosen inputs. While the setting is idealized, we emphasize that this is the first work to provide rigorous theoretical evidence establishing the benefit of active data collection in operator learning. Extending these ideas to *nonlinear* operators is an important future direction to capture real-world settings, and we hope our work lays the foundation for further works in this area.
>
>
>
>
>
> * Since our theoretical guarantees apply only to linear operators, we focused on standard linear PDEs and did not apply our estimator to nonlinear cases.
>
>
>
>
> *  **“What are the set of functions the proposed method uses to approximate the original function?”** During implementation, our method does not require searching over a pre-specified function class via optimization. That said, as discussed in Appendix A.1, the resulting estimator can be viewed as a solution to a least-squares problem over the space of linear operators, with a specific choice of pseudoinverse.
>
>
>
> * **"Does the grid size matter in the evaluation?”**  We observed that our active estimator consistently outperforms the passive baseline across different grid sizes. The choice of $64 \times 64$ was made primarily for computational efficiency, as we trained 16 separate FNO models for different sample sizes to generate the convergence plots.
>
>
> * **“Is there any baseline for comparison in the active learning field that can be used directly for operator learning?”**  Currently, there is no widely accepted active learning baseline for operator learning. Unlike the passive setting, where empirical risk minimization on i.i.d. samples is standard, active learning strategies are typically problem-specific and do not scale well to the infinite-dimensional setting.  For example, uncertainty sampling and Bayesian algorithms will both require careful extensions to infinite dimensions.

---

### Decision · Program_Chairs · 2025-05-01

**Decision:**

Accept (spotlight poster)

**Comment:**

This paper studies the mathematical theory of learning linear operators via active learning. Bound on the excess generalization error is derived in terms of the eigenvalues of the input process covariance operator. Lower bound for passive learning is also derived. Only the proof for Theorem 3.1 in Appendix A was checked by Reviewer rbqN. Minor concerns raised focus on insufficient numerical experiments (Reviews XHLU, UK7y, fV81, E2pE), insufficient details of experiments (Reviews XHLU), incremental theoretical contribution (Review fV81), and theory relies on knowing the eigenvalues of covariance kernel (Review 1DAM).